# Hydrogen Conversion in Nanocages

Ernest Ilisca

Laboratoire Matériaux et Phénomènes Quantiques, Université de Paris, CNRS, F-75013 Paris, France; ernest.ilisca@gmail.com

**Abstract:** Hydrogen molecules exist in the form of two distinct isomers that can be interconverted by physical catalysis. These ortho and para forms have different thermodynamical properties. Over the last century, the catalysts developed to convert hydrogen from one form to another, in laboratories and industries, were magnetic and the interpretations relied on magnetic dipolar interactions. The variety concentration of a sample and the conversion rates induced by a catalytic action were mostly measured by thermal methods related to the diffusion of the o-p reaction heat. At the turning of the new century, the nature of the studied catalysts and the type of measures and motivations completely changed. Catalysts investigated now are non-magnetic and new spectroscopic measurements have been developed. After a fast survey of the past studies, the review details the spectroscopic methods, emphasizing their originalities, performances and refinements: how Infra-Red measurements characterize the catalytic sites and follow the conversion in real-time, Ultra-Violet irradiations explore the electronic nature of the reaction and hyper-frequencies driving the nuclear spins. The new catalysts, metallic or insulating, are detailed to display the operating electronic structure. New electromagnetic mechanisms, involving energy and momenta transfers, are discovered providing a classification frame for the newly observed reactions.

**Keywords:** molecular spectroscopy; nanocages; electronic excitations; nuclear magnetism

The intertwining of quantum, spectroscopic and thermodynamical properties of the hydrogen molecule led to the discovery in 1927 that the proton is a fermion of spin $1/2$. It unified the nuclear spin alternation of the Lyman bands with the differences in the specific heat by considering the gases with odd and even rotational quantum number as two separate gases which do not interconvert, the ortho and para varieties. Their lifetimes are longer than the age of the universe for isolated molecules but are observed to be quite short when the hydrogen interacts with a solid. Their ranges of orders, scale from seconds when interacting with magnetic solids, to minutes for noble metals, hours for semiconductors or days for dielectric nanocages. These lifetimes recently measured are new hyperfine measures of the nuclear symmetry breakings in the hydrogen molecular space.

Although the molecular hydrogen nuclear spin isomers are well known and have been extensively studied for a century [1–3], new methods and new materials have brought an important renewal of their interconversion and thermal accommodation properties with a solid catalyst. There are already a few reviews [4–6] that report these properties and their renewals. However, the fast changes occurring now in the hydrogen research, technological and economic challenges necessitate an up-to-date relation between a synthetic summary of the present knowledge and an oriented perspective of the hydrogen catalysis. The following prospecting review extends first the conceptual notion of a molecule adsorbed in front of a solid site to a molecule enclosed inside an electric cage. That notion of a cage is relative to an electronic structure in which the molecule is imprisoned for a while either vibrating or rattling back and forth and tentatively escaping. Such a point of view enlarges the concept of physisorption beyond the particular equilibrium state. New procedures and new materials might force the molecules inside the pores (of nano-size) or dilute them inside a crystal structure (of angstrom size) or retain them in a viscous cage of a liquid solvent or a polymer, or inside a compound structure, and the molecule reacts in trying to

accommodate thermodynamically to the electric cage, since the sampled hydrogen is not at equilibrium.

Another extension that has emerged in the consideration of the conversion processes is that the hyperfine interactions and measurements are not purely nuclear processes but involve also electronic ground and excited states. New adsorption patterns, intermediate between a physical and chemical adsorption (for which the term "meta-sorption" might be used), are characterized by strong isosteric heats or forced situations in which the molecule keeps some mobility although exchanging electronic charges, momenta and energies continuously.

The following reviews the variety of optical and electronic devices, measurements and interpretations that have been reported since the turning of the new century. It is a tentative move to relate two concepts: the electromagnetic catalytic drift with the radiative measure, both being treated qualitatively with few formulas, calculations and measurement procedures.

The exposition is progressive. I review first the knowledge accumulated on the hydrogen conversion over the past century, then the new measurement methods that were operating over the last twenty years. In Section 3, I detail the new experimental devices and the new materials that have changed the catalytic properties of molecular hydrogen. Section 4 is then devoted to the recent renewal of the theoretical concepts that interpret these new observations. It focuses on the hyperfine observations of the molecule–solid exchanges and stresses the collective character of the catalytic process. Section 5 outlines a few research directions and a few industrial applications.

## 1. Hydrogen Conversion in the XX th Century

### 1.1. Molecular Symmetries in the Hydrogen Configuration Space

The electron and proton fermion characters are linked in the Molecular Hydrogen Configuration space. They build the specific symmetries of the hydrogen molecule. In the electronic ground state, the nuclear spin–rotation manifolds twine the quantum, spectroscopic and thermodynamical properties of the molecular hydrogen gas. The ortho (o) and para (p) varieties might be considered as independent gases but only special proportions are in thermal equilibrium with the environment. For most o-p mixtures in contact with a thermal bath, irreversible processes are driving the mixture. Hyperfine catalytic effects recently discovered have renewed the measurement methods of hydrogen conversion and their theoretical interpretation [5,6].

The existence of the nuclear spin isomers—ortho (o) and para (p) varieties of molecular hydrogen—illustrates the Pauli Principle: the total wave functions of molecular hydrogen must belong to the alternate representation of the permutation group of the nuclei (and similarly by permutation of the electrons). Inverting the position of the nuclei also changes the electron-nuclei relative positions. Consequently, nuclear spin and rotational parities are associated owing to the parity of the electron state. In the fundamental $X\ ^1\Sigma_g^+$ state (where the inversion of electrons is gerade and the reflection through a vertical plane positive), the spin and rotational states have opposite parities. J and I are the rotational and spin nuclear momenta; the total nuclear spin I resulting from the angular addition of the two nuclear half-integer spins.

The lower part of the Hydrogen nuclear {I–J} spectrum is represented in Figure 1. At room temperature, the non-magnetic (I = 0) para manifold (antiparallel nuclear spin configuration) is split in two different rotational components: J = 0 and J = 2, about 600 cm$^{-1}$ apart. A para $H_2$ mixture equilibrated at 300 K ($o_e = 3/4$; $p_e = 1/4$) contains a proportion of molecules of about half–half in the rotational states J = 0 and J = 2 (respective populations at T = 300 K: $p_{J=0} \cong 0.513$ and $p_{J=2} \cong 0.47$). The ortho manifold is magnetic and characterized by a "parallel" nuclear spin configuration. Although the total nuclear spin manifolds, I = 0 or 1, result from the angular addition of the two protons half-integer spins: $|(1/2,1/2)Im_i\rangle$, their energy differences are related to the overall molecular rotation and fall in the far

infra-red region. In particular the first two ortho and para states (J = I = 1) and (J = I = 0) are about 118 cm$^{-1}$ = 14.7 meV apart.

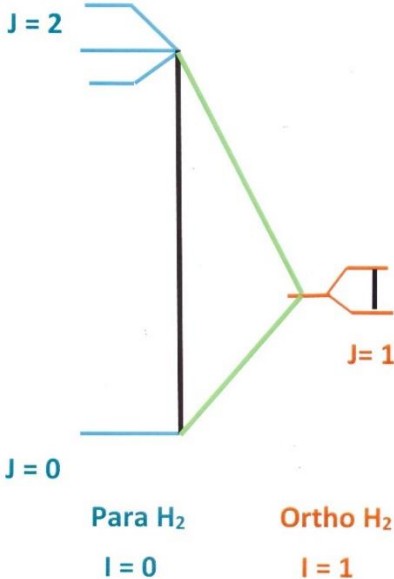

**Figure 1.** Molecular Hydrogen Spin-Rotational Spectrum. The two quantum numbers I = 0 and I = 1 characterize the two different varieties para and ortho. The first two ortho and para states (J = I = 1) and (J = I = 0) are about 118 cm$^{-1}$ = 14.7 meV apart. The nuclear spin triplet (I = 1) of a single ortho manifold (J = 1) is surrounded by a pair of spin singlet (I = 0) para ones, of lower (J = 0) and higher rotational energies (J = 2) about 600 cm$^{-1}$ apart. Green lines represent ortho-para transitions whereas black ones are the fast thermal bath transitions. (The weak proportion of molecules in the J = 3 state of population ≪ 0.09 is neglected).

Hydrogen conversion (transition between the ortho and para manifolds) is neither spontaneous (ortho or para lifetimes of isolated molecules are comparable to the age of the universe) nor induced by some radiation (the hydrogen molecule has no electric dipole moment). It necessitates a catalyst to break simultaneously the two selection rules: a double change of the nuclear spin and rotational angular momenta parities. Interactions with a catalyst relax the hydrogen towards equilibrium in times whose orders of magnitude are centuries in the diluted atmosphere, months in gases at room temperature and standard pressure, days or hours in compressed gases, or when diluted in insulating or semi-conductor cages, minutes or even shorter when interacting with efficient catalysts. Experimentally, the conversion process is observed as an irreversible relaxation of a hydrogen mixture, enriched in one isomer, towards its equilibrium concentration.

Note that the nuclear coupling is opposite for the odd (and +) electronic terms: (J = 0, I = 1) and (J = 1, I = 0). In that way, electron and nuclear symmetries (space and spin) of the ortho and para states are intricately linked. In particular, the hyperfine interactions incorporate some electron magnetism inside the ground state.

Coming back to the ground state $^1\Sigma_g^+$, a rather usual approximation consists in describing the lowest levels of the vibrational spectrum by the harmonic inter-nuclear oscillation and the molecular rotation by the rigid rotor model. IR spectroscopy probes excitations in which $H_2$ undergoes vibrational transitions that conserve the nuclear spin but might involve a change in the rotational quantum number. It is common to denote by Q(J) the purely vibrational transitions (ΔJ = 0) and by S(J) the rovibrational ones (ΔJ = 2). Although an isolated $H_2$ molecule is IR inactive, its interaction with neighboring atoms activate these transitions, shift their frequencies and often cause band splittings. When the rotation is free the eigen-energies $\varepsilon_j = BJ(J + 1)$ are degenerate [$g_{IJ} = g_I(2J + 1)$, $g_{I=1} = 3$, $g_{I=0} = 1$] and associated to the spherical harmonics eigen-functions $Y^J_M$. (B = 59 cm$^{-1}$ in the ground vibrational sate and 56 cm$^{-1}$ in the first excited one).

### 1.2. Thermal Properties of the Rotational System

The nuclear space is divided into two subspaces corresponding to the two ortho and para manifolds v = {o, p}, they manifest their properties as two different isomers (or varieties) and in some respects as two different gases of different specific heats and different nuclear magnetism.

Analyses of o-p mixtures are still often measured by thermal conductivity, thermo-resistivity or by NMR, for instance, the heat given up by an electrically heated wire stretched up in a cylindrical cell containing the gas, while the resistance of the wire is simply measured by a Wheatstone bridge. The separation of the two varieties is realized by chromatography, and for some mixtures by accommodation with a thermal bath or a catalytic action. Let me examine now the concomitance of two characteristic effects in a sample of molecular hydrogen: (i) a mixture of two hydrogen gases becoming a single one (ii) the transition from a separate equilibrium of the two hydrogen gases with a thermal bath to a global equilibrium one.

#### 1.2.1. Partitions, Populations and Energies

Let us first consider a sample of hydrogen in thermal equilibrium with a reservoir at the temperature T. The partition function $Z_J$ of any state J of degeneracy $g_{I,J}$ and energy $\varepsilon_J$ is obtained from the Boltzmann statistical distribution: $Z_J = g_{I,J}e^{-\frac{\varepsilon_J}{kT}} = g_{I,J}e^{-\beta\varepsilon_J}$, where if $\varepsilon_J$ is expressed in cal/mol, R = 1.988 cal. $mol^{-1}$ $K^{-1}$ $\left(\beta = \frac{1}{RT}\right)$ and M = 2.09 g/mol. Each population of the degenerate states J is then given by: $n_J(T) = Z_J/Z$ where the total partition function is the sum $Z = \sum_J Z_J$. The populations of each manifold, defined by the nuclear spin character, are either the total ortho concentration: $o(T) = \sum_{J \text{ odd}} n_J$ or the para one: $p(T) = \sum_{J \text{ even}} n_J$, taking into account the total concentration: o + p = 1.

The rotational space is divided in two manifolds corresponding to the ortho and para varieties v = {o, p}. Their partition functions $Z_v = \{Z_o, Z_p\}$ are defined by: $Z_v = \sum_{J \in V} Z_J$.

The ortho and para populations are then given by: $v(T) = \frac{Z_V(T)}{Z(T)}$ with $Z(T) = \sum_V Z_V$. The equilibrium ratio: $\rho(T) = o(T)/p(T)$ is used by the experimentalists to check if the sample has reached a full equilibrium. Quite generally, it seems useful to define the ortho-para population ratio as a parameter that characterizes the proportion of the two manifolds $\rho = o/p$. Each manifold reaches a separate thermal equilibrium in a time very short compared to the full relaxation time necessary to equilibrate the mixture. Non-equilibrium mixtures can be prepared either by changing the temperature of the thermal bath or by chromatographic separation. The mixtures commonly prepared by a thermal method are for example the normal hydrogen n: $\rho(n) = 3 \cong \rho$ (300 K), or the pure para p: $\rho(p) = 0 \cong \rho$ (0 K), or the half–half mixture h: $\rho(h) = 1 \cong \rho$ (77 K). For larger ortho enrichment, as mixtures prepared by chromatography $\rho$ can be large, in particular for pure orthohydrogen: $\rho$ (o)→∞.

If the manifold V is experimentally prepared at the relative concentration $\rho$, with populations $v(\rho) = \{p(\rho), o(\rho)\}$, without any interconversion, the two manifolds behave as two distinct gases. If each manifold is equilibrated with a thermostat at T but not one another, the population of a state J belonging to the variety V in the mixture of proportion $\rho$ can be written as: $n_J(\rho, T) = n(J \in V, \rho, T) = v(\rho) \varsigma_J(T) = n_J(T)\frac{v(\rho)}{v(T)}$. $v(\rho)$ is either the total ortho concentration $o(\rho)$ or the para one $p(\rho)$ at the proportion $\rho = \frac{o(\rho)}{p(\rho)}$ and $\varsigma_J(T) = \frac{Z_J}{Z_v} = \frac{n_J(T)}{v(T)}$ is the population of the equilibrated variety V that satisfies $\sum_{J \in V} \varsigma_J = 1$ and $\sum_{J \in V} n(J, \rho, T) = v(\rho)$ and: o + p = 1.

At global equilibrium of temperature T, the two varieties are sub-manifolds of the same gas. When $\rho \to \rho(T)$ each state J of manifold V recovers the population: $n_J(\rho(T), T) = n_J(T) = v(T) \varsigma_J(T) = \frac{Z_J(T)}{Z(T)}$.

When the manifolds V are separately equilibrated at the temperature T, with global populations $v(\rho)$, the populations of every J states $\in$ variety V have thermal ratios at T: $\varsigma_J(T) = Z_J/Z_v$, the effective energies of the J states $\in$ variety V: $\varepsilon_j(T) = \varsigma_J(T)\,\varepsilon_j$ lead to a total energy of the variety V: $E_V(T) = \sum_{J \in V} \varepsilon_J(T)$. In that case the energy of the mixture is obtained as a linear function of the separate energies: $E(\rho,T) = \sum_V v(\rho)\,E_V(T)$. At the limit T: $n_J(T) = z_J(T)/Z(T)$, the global equilibrium is recovered: $E(T,T) = RT^2 \sum_V v(T)\,\partial_T \ln Z_V(T) = RT^2 \partial_T \ln Z(T) = E(T)$.

The rotational specific heat of a mixture is a linear combination of the individual specific heats of the varieties V at separate equilibria: $C(\rho, T) = \sum_V v(\rho)\,C_V(T)$, where each variety has its own specific heat: $C_V(T) = R\,T\partial_{TT}^2 \ln Z_v(T)$. However as the mixture converges towards equilibrium at T, $\rho \to \rho(T)$, $\rho$ is changing continuously. The limit reached by C when $\rho \to \rho(T)$ differs from the equilibrium one: $C(T) \neq C(\rho(T), T)$. In general the specific heat at global equilibrium: $C(T) = R\left\{T^2\partial_{TT}^2 + 2T\partial_T\right\}\ln Z(T)$, cannot be expressed as a linear combination of the individual-specific heats.

Such a discontinuity was first underlined by Dennison [1] who noticed that the rotational specific heat of a gas in which o-p hydrogen equilibrium is momentarily established is completely different from the one in which a mixture is prepared with an unchanged o/p ratio such as the normal one. The specific heat of an e-mixture, for example, is maximum at low temperature (around T = 50 K C $\cong$ 4 cal/mole), whereas at that temperature the individual specific heats are vanishingly small ($C_v(T) \leq 4 \times 10^{-2}$ cal/mole). The specific heat of an e-mixture converges towards a linear combination of the individual-specific heats only above 150 K and in particular towards the normal one. The discontinuity: $\Delta C = C(\rho(T), T) - C(T) = o(T)C_o(T) + p(T)C_p(T) - C(T) = R\beta^2\,\partial_{\beta\beta}^2 \sum_V v(T)\ln v(T)$ can be related to the irreversible flow and to entropy creation.

### 1.2.2. Rotational Entropies

It is interesting to underline that the irreversible return to a thermodynamical equilibrium changes the o/p proportion and thus the mixing entropy. Consider an o-p mixture in complete equilibrium with a reservoir at the temperature T. In an infinitesimal variation of temperature dT (towards a new equilibrium T + dT), the entropy received by the system in such a reversible process is: $dS(T) = dE(T)/T = CdT/T = R\,d\{\partial_T T \ln Z(T)\}$. The total entropy of the rotational system, defined by the state function: $S(T) = \partial_T RT \ln Z(T) = R \ln Z(T) + E(T)/T$, characterizes the system equilibrium at the temperature T.

Consider now an o-p mixture prepared at a temperature t in a proportion $\rho$ and brought into contact with a reservoir at the temperature T with unchanged $\rho$. For an infinitesimal variation dT, the system receives the energy: $dE(t,T) = \sum_V v(t)dE_V(T)$, and the entropy received from the thermostat in the reversible path is obtained as: $dS^r(\rho, T) = \sum_V v(\rho)\,dS_V(T)$; with: $dS_V(T) = \frac{R}{T}\,d\{T^2\partial_T \ln Z_V\}$. An entropy can thus be attributed to each variety, equilibrated separately at the temperature T: $S_V(T) = R\,\partial_T T \ln Z_V(T)$ and the total entropy of the mixture is the sum of the individual entropies: $S^r(\rho, T) = \sum_V v(\rho)\,S_V(T)$. However, if the o-p mixture, prepared in a proportion $\rho$, is being brought into complete equilibrium with a reservoir at the temperature T, an additional entropy is created which is associated with the conversion process. Such an irreversible path from the initial preparation of a mixture where the two varieties have been equilibrated separately at temperature T in a proportion $\rho$, towards a global equilibrium at T creates the additional entropy: $S^c(\rho, T) = S(T) - S^r(\rho, T) = R\,\partial_T T.\left\{\ln Z - \sum_V v(\rho) \ln Z_V\right\}$. Considering that each variety, ortho or para creates its own entropy when its population changes of magnitude: $S_V^c(T) = -R\,\partial_T T. \ln v(T)$, the whole mixture adds the balanced sum of each

created variety entropy: $S^c(t, T) = \sum_V v(\rho) S^c_V(T)$. Since such an irreversible relaxation of the molecular system towards its global equilibrium corresponds to the molecular conversion, that created entropy can be called the conversion entropy: $\Delta S_{conv}(\rho, T) = S^c(t, T) = -R \, \partial_T T . \sum_V v(t) \ln v(T)$.

Finally, consider the limit $t \to T$, where the initial preparation approaches the equilibrium one at T. The created entropy: $S^c(T) = \lim_{t \to T} S^c(\rho(t), T) = S(T) - S^r(T, T)$, leads to the function: $S^c(T) = -R \sum_V v(T) \ln v(T) = S_{mixing}$, which is precisely the entropy mixing associated to a particular distribution of the molecules between the varieties. It is positive: $S^c(T) > 0$, because $v(T) < 1$, and that character is associated with the irreversible path towards equilibrium. When flowing towards a global equilibrium the variation of the o-p ratio changes the number of indiscernible particles in each variety, and creates an additional entropy. The hydrogen molecules have the peculiar character to change from discernible to indiscernible (different or identical nuclear spin) when their manifolds equilibrate each other.

### 1.3. From Experimental Studies to Industrial Applications

The book of A. Farkas [1] published in 1935, still an actual reference, formalized the distinction between the two families of conversion mechanisms, namely the chemical and the physical ones. The chemical conversion operates by a temporary molecular dissociation, followed by subsequent recombination along statistical proportions, in accordance with the thermal conditions of the interacting systems.

The physical conversion is quite different, it relies on the electromagnetic interactions with a catalyst assuming no chemical bonding. That physical model elaborated in 1933 by E.P. Wigner [7] relies on the integrity of the $H_2$ molecular electronic structure and is based on the inhomogeneous magnetic field produced by a magnetic moment able to uncouple the nuclear spins of a nearby hydrogen molecule and diphase of the nuclear spin precession of the two protons. Wigner's theory assumes an instantaneous symmetry-breaking collision with a paramagnetic catalytic site in a time function of the sample temperature. Shortly after the discovery of the hydrogen spin isomers, Wigner's theory has allowed the interpretation of a large variety of experimental measures, based on a conversion rate proportional to the squared of the magnetic moment and the inverse of the sixth power of the molecule-catalyst site distance. Dynamically, the electro-nuclear spin systems interacting through dipolar forces transfer angular momenta and energy between the molecular rotational ones and the catalyst.

In the sixties, the ortho-para proportion was still measured by thermal methods based on their different specific heats and related thermodynamical properties (for instance the heat given up by an electrically heated wire in a cell filled with hydrogen) and up to now, hydrogen is practically converted to thermodynamic proportions, by passing through magnetic catalysts. NMR was effective to measure the spin relaxation in $H_2$ gases [8] but gave little information on the conversion processes. Only seldom theoretical approaches appeared. Nielsen and Dahler in 1967 based their approach on the binary collisions of the magnetic catalyst and reactant species using the distorted wave approximation [9] but their calculated rates turn out to be weaker than the experimental ones. Many studies in 1950–1970 tried to improve the local magnetic gradients able to break the spin and rotational nuclear selection rules to speed the conversion efficiency.

When I started in 1968 to investigate the conversion dynamics in the low-temperature group of the Ohio State University, the ferromagnetic character of most efficient catalysts was mysterious and considered as a parasite effect, mostly because the surface magnetic fields of ferromagnetic samples being rather homogeneous on a microscopic scale would be unfavorable for the catalyzed conversion. The main experimental patterns should be first summarized.

In between 1954 and 1959 the US National Bureau of Standards laboratories performed extensive programs to establish the best conversion catalysts. The commercial, military and

aerospace interests were to supply the rocket engine testing programs and the hydrogen-fueled rocket vehicles being developed by the Air Force and by NASA. Various transition metallic oxides such as manganese, ferric, chromium and more in the form of powders or beds were studied [10]. The unsupported hydrous iron oxides proved to be the most effective. Although the increased activity of the iron oxides over the chromium ones was expected, the measured improvement was many times greater than expected. These hydrous iron oxides which exhibited ferromagnetism were the very samples that showed the highest activity, about 5 times more effective than the chromium oxide ones. A similar pattern was also reported on $Fe_2O_3$ samples, although on a weight basis the 0.5% Ni-Alumina was even more active.

On the Russian side, Buyanov in 1960 selected hydroxides gels of highly developed surface area 100–300 $m^2/g$, with $Cr^{3+}$, $Mn^{4+}$, $Fe^{3+}$, $Co^{3+}$, $Ni^{2+}$ of have partly filled 3d shells. The best catalysts found were in decreasing order Cr-Ni, $Fe(OH)_3$, $Mn(OH)_4$, $Cr(OH)_3$, $Ni(OH)_2$, $Co(OH)_3$ [11]. It must be stated that no chemical substance of these formulae is listed in the Magneto-Chemistry reference books, and the exact compound mixtures of the best effective catalysts still remain confidential.

From 1962, extensive studies of the magnetic catalysts were performed by the Californian team of Pr. P.W. Selwood [12–14]. First on alumina supported oxides, and taking the rate observed on commercial chromia-alumina as a reference, the supported 0.5% NiO, 0.5% Ni, and 5% Ni were respectively 5, 53, 90 times more effective. Puzzled by the contradictory effects of exchange interactions in the catalysts, the authors asked: "Is there any reason why the presence of ferromagnetism should be associated with catalytic activity?" but concluded, despite their troubling results that these observations "are attributed to accidental factors not related to ferromagnetism". However, Selwood reported measurements on magnetically concentrated catalysts, extensively from 1969. The most remarkable results concern antiferromagnetic samples. Chromia, Cobalt monoxide and manganese monoxide were investigated. The o-p conversion rate on chromia displayed, in the function of temperature, an abrupt change at the Neel transition temperature of 308 K, as represented in Figure 2. Such a pattern is representative of the famous "Hedvall" effect.

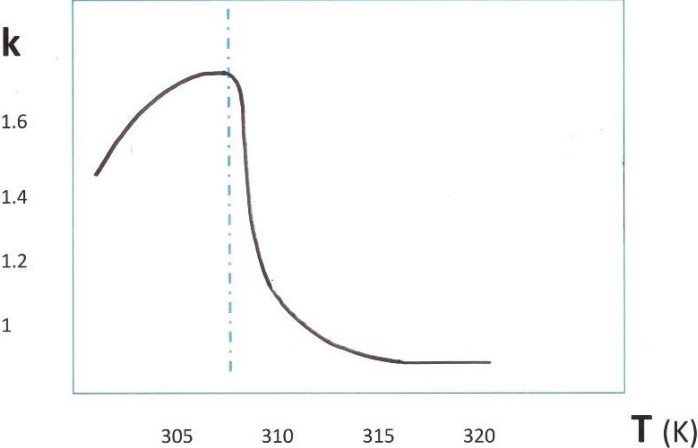

**Figure 2.** Hydrogen Conversion Rate Discontinuity at the Neel Temperature. The conversion rates k expressed in $mol \cdot m^{-2} \cdot s^{-1} . 10^4$ versus the temperature T (in K) over $Cr_2O_3$ in the close vicinity of the transition Neel temperature (indicated by a dashed vertical line).

Hydrogen conversion became an industrial challenge because almost all liquefied and stored hydrogen has to be converted (around $10^{10}$ kg/year, a major part of which for aerospace rocket and engines). Undoubtedly, the subject has been dominated for a long time by industrial concerns. The need to feed the combustion motors to propel the American rockets involved the production and the liquefaction of huge quantities of hydrogen. Then the fact that the o-p energy released in the conversion (168 cal/g) being larger than the vaporization heat (108 cal/g) obliged the producers to convert the liquefied

hydrogen into its lower para form to store and prevent its evaporation and consequent losses (about 30% would evaporate per day if not converted). Experimental research was thus directed towards the study and production of efficient catalysts, interacting with the largest number of molecules and thus made of porous structures, irregular surfaces dispersing important amounts of magnetic impurities. The resulting effect of so many physical phenomena was condensed in one observable mean value: the catalytic rate and measured by one thermal heat, that released by the catalyzed conversion [2,3,10,15].

*1.4. First Theoretical Models*

Magnetic Conversion has since 1933 denoted the hydrogen conversion induced by a first-order magnetic interaction between a hydrogen molecule and the magnetic moment of a localized ionic impurity. The dipolar processes give a simple interpretation of the conversion measurements on a large variety of magnetic substrates. The original theoretical approach given by Wigner in 1933 demonstrated that the strongly inhomogeneous magnetic field of paramagnetic impurities decouples the proton spins of the hydrogen molecule [7]. They are thus able to break the double electronuclear symmetry link. All the experimental results reported for about 50 years, between 1933 and 1983, have been interpreted by the Wigner law: "conversion rate k $\sim$ $\mu^2 d^{-6}$" in the approximation of sudden collisions with a localized magnetic moment $\mu$ at a molecule-magnetic impurity distance: d. Wigner theory" was established for a hydrogen gas in which magnetic impurities (either atomic, molecular or ionic) were diluted. The true paramagnetic Wigner conversion rate expression is: "k $\sim$ $\mu^2 d^{-8}$" but it was modified to introduce a sudden interaction occurring in the time t$\sim$d/v, where v is the thermal molecular speed in the gas, transforming thus the law in: k$\sim$d$^{-6}$. When the Wigner model was applied to a solid catalyst surface, experimentalists had difficulties in checking the rate–distance dependence and defining the signification that should be attributed to the distance d [5,7].

I gave the first microscopic theoretical treatment of the conversion rate beyond Wigner's approximation of sudden collisions in 1970 and 1972 [16–20]. The theory was provided for hydrogen conversion occurring inside the surface adsorbed layer of para-magnetic and ferromagnetic catalysts. Let me concentrate first on the paramagnetic conversion model. More details on the magnetic catalytic processes will be discussed in Section 4.1.1. Quite generally, the $H_2$ conversion rates were expressed by products of quantum couplings and spectral densities at the $H_2$ o-p frequencies. The quantum part was obtained by expressing the matrix elements of the dipolar interaction among the three systems: molecule nuclear rotation and spin, and electron spins of the catalyst. The spectral density was written as the Fourier transform of the correlation functions of the molecular displacements at the surface (treated as classical variables) and of the electron spin dynamics. The various dynamical motions at the catalyst surface are schematically represented in Figure 3. Altogether, the ortho-para conversion rate was calculated with the following five assumptions:

(i) At low temperatures, T $\leq$ 100 K, only the two lowest ortho and para ground states are populated (at T = 100 K about 99% of the molecules are in the J = 0 and J = 1 states)

(ii) When a hydrogen molecule is adsorbed, its rotational motion is seriously altered by the surface, and behaves approximately as a plane rotator parallel to the surface.

(iii) The surface of the catalyst is planar on the scale of the molecular travels. A few cases of physical adsorption were investigated: The almost filled adsorbed monolayer behaves as a two-dimensional ideal gas, (described by a two-dimensional diffusion equation); or the adsorption is localized and each molecule jumps from one site to another in a random walk.

(iv) When the molecule receives enough energy from the solid (phonons) or from the gas (collision with another molecule), it leaves the surface.

(v) Magnetic impurities are randomly dispersed on the surface and their isotropic relaxation is described by an exponential decrease.

Thereafter, Petzinger and Scalapino enlarged in 1973 [21] my paramagnetic formulation of the conversion rates in studying the temperature dependences of a few different molecular motions at the catalyst surface: diffusions at the surface and successive jumps. Atkins and Clugston extended in 1974 [22] the previous rate calculations to the case of solutions where paramagnetic ions are diluted in liquids. Later in the eighties, I extended the quantum formulation of the conversion process using a density operator formalism, to include non-diagonal effects, and applied the formalism to interpret the measurements of the conversion speed on Chromia, Nickel and various 3d magnetic ions [23–25]. In the nineties, I interpreted also the magnetic field effects observed by P.W. Selwood, on the basis of the electron spin-orbit fine structure [26,27].

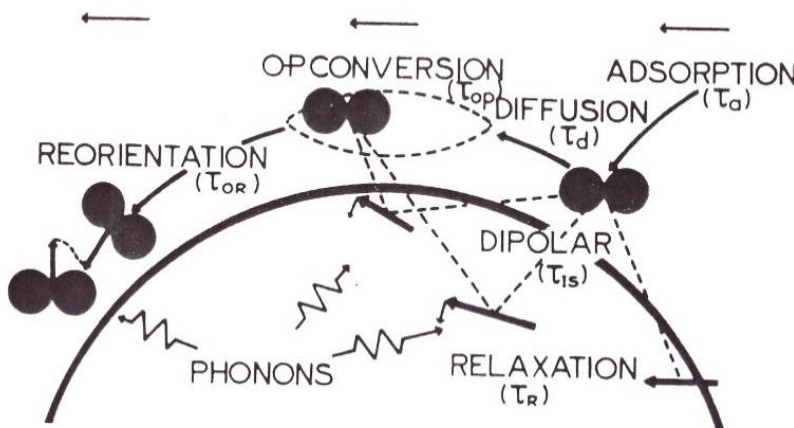

**Figure 3.** Hydrogen Adsorption on Paramagnetic Surfaces. Molecule–Solid Dynamical Interactions with a Thermal Bath: Diffusions (Translational and Rotational)—Dipolar Relaxations (electronic and nuclear). The solid phonons thermalize both the molecular adsorbed layer as well as the magnetic spins. The various correlation times that drive the system dynamics are indicated. Reprinted with permission from [23] as follows: Ilisca, E., Phys. Rev. Lett., 1978, 40, 1535, Copyright (1978) by the American Physical Society.

## 2. The New Conversion Measures of the XXI th Century

Since 1980, numerous diffraction studies of molecule, electron or neutron beams scattering a catalyst surface were performed [28–31]. Surprisingly, a quite fast conversion was observed in 1982 on non-magnetic noble metals, separately at IBM for silver samples [32] and at Chalmers for copper samples [33], and later on graphite [34]. Such observations performed by EELS (Low Energy Electron Spectroscopy) were very challenging since the o-p transition: $\Delta J = \Delta I = 1$ was supposed to be strongly forbidden in absence of a nearby magnetic moment (it requires a simultaneous parity change of the rotational and nuclear spin momenta). Although the experiments by EELS could not measure precise conversion rates, these experiments have established the primary observation of conversion on (i) non-magnetic catalysts (ii) metals without chemisorption (iii) single crystals covered by freely rotating hydrogen molecules. My interpretation of these measurements published in 1991 [35–40] and described in Section 4.1.2 was followed by the experimental confirmation performed by the Tokyo ISS Fukutani's group, in 2003 [41,42]. It opened a new branch of the hydrogen conversion history: the non-magnetic physical catalysis of molecular hydrogen.

Important changes have then occurred at the turning of the century in the experimental measurements of the nuclear conversion of hydrogen gases. Hydrogen conversion is now investigated by a variety of optical and electronic devices [43–52] on a variety of non-magnetic catalysts.

The fact that these opto-electronic measurements were registered on non-magnetic solids (a few tenths of experiments) and very few on magnetic ones [53–55] proceeds from the conversion times magnitudes. Efficient magnetic catalysts, even commercial ones, affect quickly the hydrogen mixture's concentration, in times shorter than the minute. These

rates are too fast to be followed by the present spectroscopies but also by the time needed for the molecules to reach the solid samples. Inversely, the fact that o-p conversion can occur on non-magnetic solids broadened the time scale and opened a new field of research, developing new measures, leading to new information but also to yet unsolved questions.

Basically, the important characteristic that differentiates these new methods is some ability to observe the dynamical evolution of the hydrogen varieties. Old thermal measures are more static, they register the ortho-para relative concentrations. The new optical methods are able to measure the conversion rates and follow the dynamical evolution of the reaction. Some of these technics observe these evolutions « in real-time », some others reproduce them with some delay. Some IR vibrational technics observe the molecules « in situ », can identify the adsorption sites and sample the feeding and adsorbing capacity of one site as well as identify the eventual motion from one site to another. UV technics necessitate extracting the molecules from their supports, before analyzing their content, but they bring important information on the electron transient states during the catalytic process. There is also a reasonable hope that they might be able to operate soon « in situ » and « in real-time ». Radio-, Hyper- and Tera-Hertz irradiations are more sensitive to the nuclear spin dynamics and have the advantage to observe the samples « in situ » and almost « in real-time ».

The experimental techniques able to distinguish ortho and para varieties of molecular hydrogen might be divided between those probing the nuclear spins and those probing the rotational states. The former can again be divided into two varieties: scattering methods such as neutron beams and radio-frequency electromagnetic irradiations such as periodic pulses or Nuclear Magnetic Resonance adapted to the ortho relaxation times. NMR has a low sensitivity on surfaces but is quite efficient in liquid and solid $H_2$. Other methods rely on the rotational spectroscopy that can be achieved either by pure rotational or vibrational or electronic excitation, whether accompanied by rovibrational excitation or not. Pure rotational or vibrational excitation might be achieved either by Raman Scattering Spectroscopy RSS, or Inelastic Neutron Scattering INS, or Electron energy loss spectroscopy EELS, or Infrared Absorption Spectroscopy IRAS. Finally, the electronic excitation probes became of increasing efficiencies, in particular the multi-photon laser methods REMPI, or those based on STM schemes for metal surfaces. Scattering beam methods such as EELS, sketched in the following chapter had an important historical role, but is unable to measure the dynamic phenomena. Raman RSS and INS have a weak sensitivity and thus are not suited for surfaces. They are however efficient for powders or porous oxides, such as carbons or MOF; they give complementary information that will be sketched further.

Two different classes of electromagnetic measurements will be described: in the first class, the hydrogen molecules are observed "in situ", when the molecules are in front of catalytic sites and when possible in real-time or eventually afterward when the molecules have been converted. Another class necessitates some delay, a preliminary step when a partial interaction is interrupted by a pulse (thermal or radiative) the molecules being separated from their catalytic partners and then the second step measures the o-p relative concentrations by radiative spectroscopy. In the following, I concentrate successively on the IRAS, REMPI and NMR methods of measurements.

## 2.1. Infra-Red Spectroscopy

Let us first examine infra-red (IR) spectroscopy, efficient for powders or porous oxides (such as MOF), carbons and semiconductors. As a light molecule, the quantum characters of the nuclear rotation appear clearly in the resolved vibration–rotation lines, since the o-p splittings are different in the excited states. Zero-point energies, rotational hindrances and lifting of degeneracies have also been explored [46]. Raman or IR excitation bands contain a great deal of information because the coupling of nuclear-spin and angular momentum couple with the arrangement of the $H_2$ molecules in the cage singularities, wherever in the pores of polymers or adsorbed in front of metal, oxygen or organic sites of the Metal-Organic Framework or diluted inside an interstitial site of a crystal. There is however a

restriction in such a measurement: the adsorbed molecules are sampled, only because of the appearance of surface-induced electric dipolar momenta (and when it appears!). The molecules have no permanent dipole moment and IRAS cannot be observed in gases. However, on solid surfaces or when interacting with ionic substrates hydrogen might be polarized, by which a dipole moment being induced, IR absorption is observed due to vibrational excitation.

Raman and infrared spectroscopies observed the hydrogen conversion from 1992 by exciting the molecular rovibronic degrees of freedom [43]. These are the first measurements of o-p hydrogen conversion rates by optical radiation. Related methods on different nuclear spin isomers of hydrogenated molecules such as $CH_3F$ and $C_2H_4$ had been employed earlier in gases [56–58], but these molecules have an electric dipolar momentum. The development of "in situ" and "site-specific" methods combined these new optical measures on a variety of diamagnetic insulators, with other surface spectroscopies such as X-rays or neutron beams [49], thermal- or photo-induced desorption. The time evolution of the rotation–vibration bands denoted: Q ($\triangle J = O$) and S($\triangle J = 2$) (both ($\triangle v = 1$)), investigated by infra-red (IR) spectroscopy in semiconductors, Metal-Organic-Frameworks and organic polymers, became one of the best tools to observe the hydrogen conversion. Hydrogen enriched in various ortho concentrations being introduced, o- and p-lines are clearly identified from their dynamical behavior. The simultaneous o-decrease and p-increase of these lines follow in real-time the isomers' relaxation towards thermal proportions.

"Site-specific" infra-red measures were reached for Hydrogen physisorbed in the pores of Metal-Organic Frameworks (MOF), by varying the amount of adsorbed hydrogen, exploring and comparing the IR lines corresponding to molecules adsorbed on different metal, oxygen or organic sites [46,48].

The low-temperature diffuse reflectance infra-red spectroscopy was used to measure the quantum dynamics of molecular hydrogen adsorbed in the micro-porous MOF samples. The diffuse reflectance apparatus conceived and realized by Stephen FitzGerald has major advantages over the transmission technics in delivering more precise and ample signals [51]. It is represented in Figure 4 and allows measurements at low-temperature, with possible variable loading of hydrogen. It provides site-specific information. In particular, for MOF-74 samples containing zinc metal ions, the hydrogen loading on the primary (1) and secondary (2) adsorption sites can be followed. These MOF sites: (1) above Zn ions and (2) above oxygen ions had already been observed and characterized by neutron diffraction, and the IR measurements add the additional possibility to correlate their conversion efficiency with their electronic distribution. It was a striking observation that the conversion rates over the metallic site and the oxygen one were of the same order of magnitude and that the loading on the secondary sites could affect the conversion rate over the first ones.

Another example of an "in situ" IR measurement, not exactly "in real-time", is given by the formation of molecular $H_2$ inside semi-conductors and in particular inside silicon samples. The microscopic behavior of molecular hydrogen in the interstitial cages remained for long a controversial issue [59,60].

It was first observed in hydrogenated amorphous silicon by NMR [61], and later by Raman spectroscopy in plasma-exposed samples [62,63]. It is stable, almost freely rotating at interstitial tetrahedral $T_d$ sites [44,63,64]. The appearance of the ortho (o) and para (p) lines in the spectra is a signature of the formation of molecular hydrogen and their intensity ratio gives access to the degree of thermal equilibrium with the host silicon. To investigate the conversion at 77 K, these samples were exposed to a hydrogen plasma at room temperature, and then stored at 77 K. The o-p ratio is then measured separately as a function of the past storage time in liquid nitrogen. At higher temperatures, after long storage at 77 K, the o-p ratio is reported vs. the annealing time at 300 K. These dynamical relaxations were astonishing and their reports have questioned for long the possibility of conversion when interacting with non-magnetic and non-metallic structures. However, these observations were extensively confirmed separately by Raman and IRAS measurements of different groups.

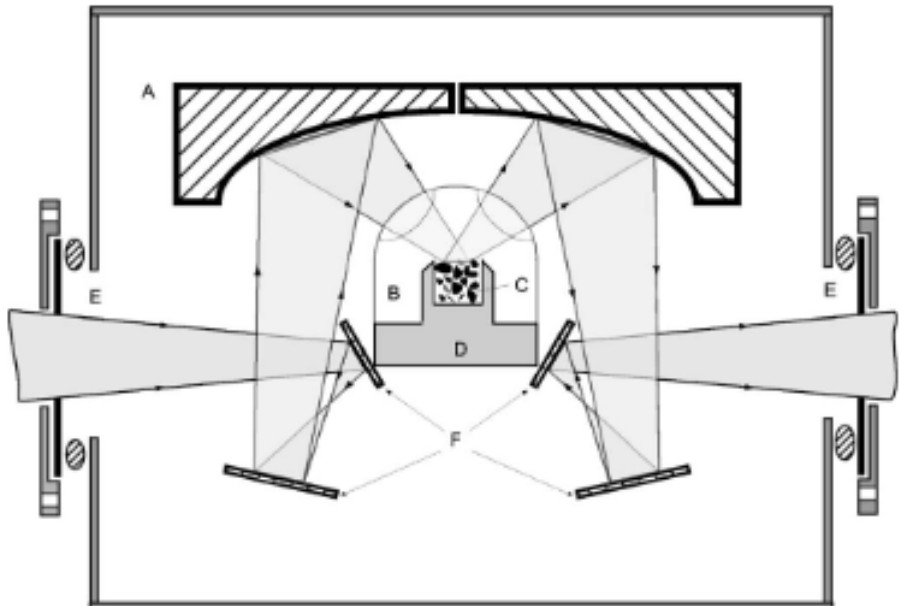

**Figure 4.** Diffuse Reflectance Infrared Spectroscopy apparatus. Schematic of radiation propagation within vacuum chamber showing the basic working of the DRIFTS optics. The incident radiation is reflected onto off-axis ellipsoidal mirrors that focus the beam through the window of the high-pressure dome and onto the sample. Within the sample the radiation undergoes multiple scattering events before reemerging at some random angle. The second ellipsoidal mirror collects up to 50% of the emerging radiation and passes it out of the chamber to be focused onto the detector via additional optics. The sample is enclosed by a high-pressure dome with optical access for infrared radiation to enter and exit. Reproduced with permission from [51] as follows: FitzGerald, S.A., et al., Rev. Sci. Instrum. 2006, 77, 093110, Copyright by the AIP Publishing.

### 2.2. UV Photo-Ionization Methods

The new Photo-Ionization method, called « Resonance -Enhanced-Multi-Photon-Ionization », REMPI, elaborated by the Tokyo Fukutani group at the beginning of the new century is different in nature from the IR ones, for two reasons: it involves a first step in desorbing the molecules before their analysis, moreover the laser measurement explores the rotational energies from the top, as measured from the vacuum energy [41,42]. Let me briefly sketch the experimental approach REMPI. (Other methods such as the Laser Induced Fluorescence LIF are described in different reviews). Figure 5 describes the transitions realized by K. Fukutani and al. together with the temporal steps of the operating procedures. In the REMPI method the hydrogen molecule is first excited through the transition X→(E, F) $^1\sum_g^+$ by two-photon absorption and then ionized by another photon [65]. The electronic excitations are able to characterize the ortho and para hydrogen states because of their different excitation energies. However, the REMPI procedure does not operate up to now directly on surfaces because of the unprecise knowledge of the molecular interactions with the solid. It has been applied only in gases and is still even precise at low pressures (up to $10^{-3}$ bar). Therefore to measure the hydrogen concentration of adsorbed molecules on solids, REMPI is combined with desorption techniques: either the Photo-stimulated Desorption PSD or the Thermal desorption TDS which eject the molecules from their adsorption sites before being probed by the laser beams. The preparation of non-equilibrium ortho-hydrogen is realized by chromatography.

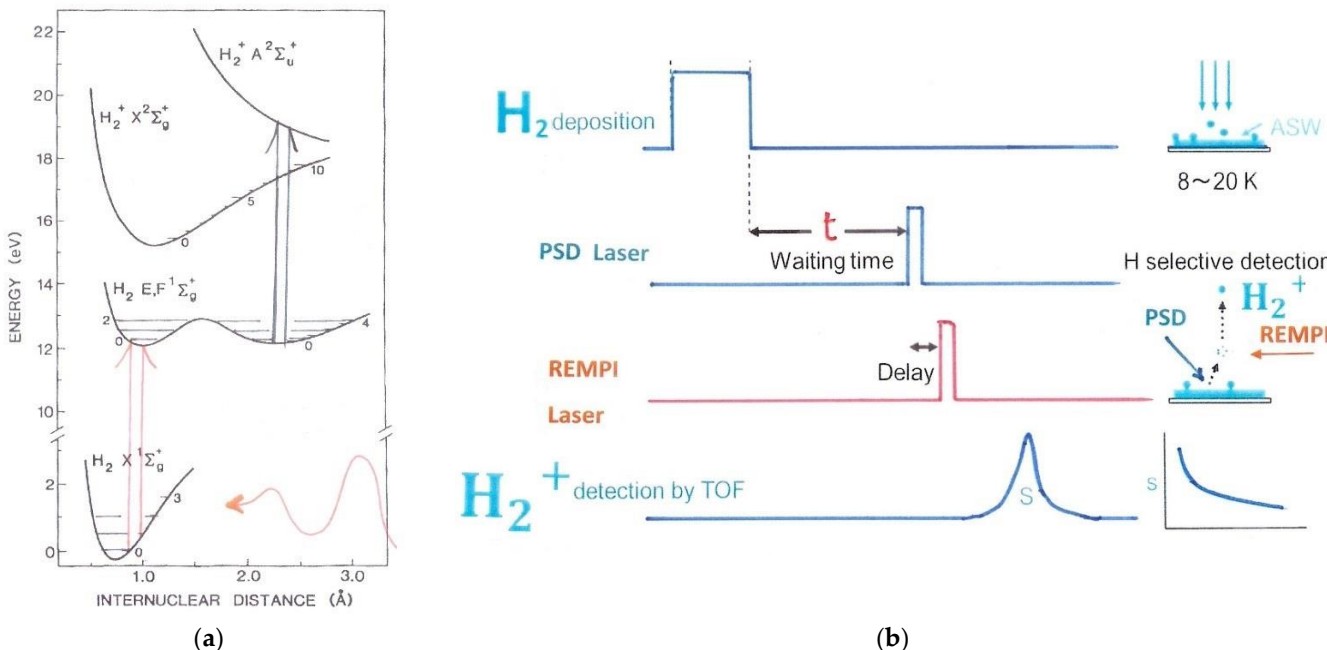

**Figure 5.** Resonance Enhanced Multi-Photon Ionization: REMPI Detection. (**a**) Laser-Induced Transitions in the hydrogen spectrum: Two-Photon transitions are the X → E,F first one; One-Photon ones are the upper and ionizing E, F → A one. (**b**) Procedure Runnings.

In 2003, the first precise experiments of hydrogen physisorbed on Ag (111) at low temperature, based on photo-desorption and molecular ionization, were realized by REMPI. The samples were prepared in UHV and exposed to $H_2$ or $D_2$. Then REMPI detects molecules upon PSD or TDS after the laser beam time-resolved analysis (REMPI) with another laser beam of 2.3–6.4 eV not shown in the figure. Desorption is identical for ortho and para varieties, but the signal is proportional to the H coverage on the surface. The conversion process could also be accelerated by increasing the laser power. Shortly later, o-p conversion of hydrogen adsorbed on amorphous ice (ASW) was also observed to be quite fast (of the order of a few minutes) [50,66]. These measurements realized major progress in the conversion history.

*2.3. Radio Frequency Pulses*

NMR spectroscopy has proven to be one of the most powerful methods in magneto-catalysis. Indeed, it is widely used for the characterization of homogeneous and heterogeneous catalysts, reactants and products in different phases, reaction intermediates, and more [67–70]. However, one of the major disadvantages of conventional NMR is its low detection sensitivity, which results from weak nuclear spin polarization under typical thermal equilibrium conditions. In order to overcome this sensitivity problem, several hyperpolarization techniques have been developed, [71,72] including parahydrogen-induced polarization (PHIP) [73–75]. PHIP effects can be observed via NMR during hydrogenation of unsaturated molecules with parahydrogen, provided that the hydrogenation process occurs via the pairwise addition of hydrogen atoms (and the nascent H positions become magnetically inequivalent). In other words, two hydrogen atoms from one parahydrogen molecule should be added to the same reactant molecule; if this condition of pairwise addition is satisfied, NMR signals of hydrogenation products and intermediates can be significantly enhanced and exhibit characteristic antiphase line-shapes [75]. This property has made PHIP a unique tool for mechanistic investigation of catalytic reactions, due to the possibility of sensitive intermediates detection and the ability to track hydrogen atoms from the same hydrogen molecule [74–76].

### 3. New Devices and New Materials

The new chapter of the conversion history occurred at the turning of the century when the rotational scattering of $H_2$ molecules interacting with a solid became measurable [28–34]. Experimental microscopic measures of hydrogen isomer concentrations benefited from a coincidence of new optical methods, new materials as catalysts and new irreversible phenomena. The main change for the o-p conversion process was that it also occurs on non-magnetic surfaces and that their measured rates could be fast! Some of them were of the order of one or a few minutes, opening thus the way for new catalytic mechanism investigations. Some others were very slow and opened new possible magnetic shielding for long conservative storage.

Conversion on non-magnetic surfaces was observed first on noble metals, then independently on ionic substrates and semi-conductor crystals, then in amorphous structures, polymers and various Metal-Organic compounds. Multiple information on the catalytic rates implied the disentanglement of what pertained to the solid and the molecular structure. On the solid side, the importance of disordered structures raised questions on a possible correlation of the conversion mechanism with the catalyst disorder.

These observations raised also questions about the influence of the catalyst ionicity and more generally on the electronic distribution all over the surrounding cage. On the side of molecular chemistry, non-equilibrium phenomena and irreversible shifts such as negative temperature effects and transient nuclear spin polarizations were observed thanks to these new electromagnetic measurements.

The motivation of these research fields relies also on the diversity of applications, as for instance: opto-electronic devices, carbon technologies, doped semiconductors conductivity, storage of the hydrogen energy, NMR medical imaging and molecular formation on the cosmic ice dust. These will be discussed in Section 5. In the present chapter, the amorphous structures are first examined, followed successively by porous catalysts, nano-carbon composites and organic polymers with various metal, oxygen or organic sites.

#### 3.1. Amorphous Catalysts

The following hydrogen catalysts, ASW and Si compounds have opposite patterns: very fast on solid water and very slow in silicon cages. Some acceleration of the conversion rate is noticed in disordered silicon samples when the molecule escapes from its cage.

#### 3.1.1. $H_2$ Adsorbed on Solid Water

The REMPI (resonance-enhanced multiphoton ionization [66]) method was initiated by K. Fukutani and its group at Tokyo University to study the $H_2$ conversion on silver samples in 2003 [41,42]. Later, in 2010, Watanabe et al., Sugimoto and Fukutani observed the o-p conversion of hydrogen adsorbed on amorphous solid water ice (ASW) by a combination of photo-stimulated desorption and "REMPI" methods [65,66,77–82]. Later on, they were able to detect hydrogen conversion on other cosmic dust analogues: polycrystalline ice (PCI), pure solid carbon monoxide, and diamond-like carbon.

The physicochemical processes of hydrogen on cosmic dust, such as the diffusion of H atoms and nuclear spin conversion of $H_2$ molecules at low temperatures, have a significant influence on the subsequent chemical evolution in space [50,77–82]. The birthplace of stars and planets, namely, the interstellar cloud consisting of gaseous species and cosmic silicate dust, are the starting point of chemical evolution in space, where most of the relatively light elements, such as hydrogen, carbon, oxygen, and nitrogen, first exist as atoms. Hydrogen plays an important role in the formation of molecules on ice mantles, which cover the silicate dust. These ice mantles of amorphous water have interesting adsorbing properties as their surfaces are characterized by giant electric fields and steep gradients. The depth of the adsorption potential on the ASW has a wide distribution unlike that on crystalline ice. The magnitude of the electric fields was shown to be between $3 \times 10^9$ V/m and $4 \times 10^{10}$ V/m at 2.35 A from a single water monolayer ice film.

The ortho-para ratio (OPR) is a tool to investigate the physical and chemical condition of the birthplace of molecules. The energy difference between the lowest para (J = 0) and ortho (J = 1) states, approximately 170 K, is significant in Molecular Clouds whose dusts are cold ∼10 K. Therefore ortho-para conversion affects not only the chemistry but also the gas dynamics of core formation leading to star formation.

The ortho-to-para conversion is characteristic of the nascent $H_2$ molecules produced by H-H recombination. The ratio (OPR) was observed to decrease from 3 over time when $H_2$ stays on the ice surface and registered to occur in between 6 and 12 min. Such a conversion time is considered very fast. In order to provide information on the conversion-rate-limiting processes the ice temperature dependence of the o-p conversion rate on ASW was measured in the range of 9.2–16 K as a function of the surface temperature. The conversion rate was found to increase steeply with the temperature and displays an important and astonishing temperature sensitivity: below 12 K the rate almost doubles for each degree increase. These data were fitted by the authors with the power law $3.2 \times 10^{-11}$ $T^{7.1}$ s$^{-1}$, a pattern that will be discussed in Section 4.2.3.

### 3.1.2. $H_2$ Diluted into Semi-Conductors

The long history and various patterns of hydrogen conversion in silicon must be briefly sketched. Hydrogen is a common and important impurity in semiconductors, which can be trapped at various sites of the host lattice [59,60]. In Si [61–63], GaAs [64] and Ge [44,45] the introduction of hydrogen in high concentrations results in the formation of extended planar defects, called platelets. It is possible to obtain interstitial $H_2$ in silicon compounds by cooling the Si from a melt in a hydrogen plasma, using either the Czochralski- or a zone-melt process. In crystalline Si hydrogenated from plasma at moderate temperatures, these structures are oriented predominantly along 111 crystallographic planes [83–87].

The "Lavrov and Weber" group in Dresden measured the nuclear o-p conversion rates in various silicon compounds by Raman Spectroscopy: single-crystalline Si with interstitial $H_2$ [45,65], bound to interstitial oxygen (O-$H_2$), trapped within Si {111}-oriented platelets and inside single Si crystals implanted with $^{28}$Si ions [88–92]. The intensity ratio of these components depends on the thermal history of the sample. Such non-magnetic conversion was challenging and considered as mysterious. However, later on, infrared measures performed by M. Stavola et al. confirmed the Raman o-p rates and patterns [64].

The following questions might be addressed: Why conversion times of tens to hundreds of hours in semiconductors and not minutes as in noble metals or ionic insulators? Why the silicon rate measured at 300 K is about 30 times faster than the one at 77 K? Why the rate in Si platelets is faster than for the natural Si one by a factor of 25 at 77 K, but only accelerated by 1.5 at 300 K?

The conversion of interstitial molecules in silicon is exceptionally slow at 77 K, (about 230 h) but as the silicon structures become more disordered in K-complexes and platelets, with gaps reduced by impurities or two-dimensionality, the conversion becomes faster (by a factor ∼25 for platelets). Contra-distinctly, the rates at 300 K remain quasi-insensitive to the gap width.

Clearly, the intermolecular interactions and molecular diffusion in two-dimensional platelets are different from the incoherent motion in nanocages and inter-site jumps. It was expected that a liquid–solid phase transition might occur at low temperature but is not confirmed until now.

Raman-scattering studies of hydrogenated Si showed that the stretch local vibrational modes « LVM » of $H_2$ trapped within platelets result in a broad band around 4150 cm$^{-1}$, very close to the free $H_2$ one. The observed $H_2$ vibrational Raman band consisting of three components were assigned: around 4150 cm$^{-1}$ to ortho $H_2$, at 4160 cm$^{-1}$ and to para $H_2$ when the molecules were trapped in platelets, whereas the broader component around 4140 cm$^{-1}$ was attributed to hydrogen molecules located in smaller voids/platelets or precursors of platelets. One of the most remarkable properties of the (111) platelets is their

two-dimensionality over diameters of many tens of nanometers while having a thickness of only a few Å.

In oxygen-rich Si samples, two of the three IR lines are associated with $H_2$ trapped near interstitial oxygen ($O_i$). Oxygen impurities have no influence at 77 K, but they speed the rate by a factor of 2 at 300 K. The highest repulsion for $H_2$ located at sites nearby interstitial oxygen atoms is consistent with the vibration frequency downshift and binding energy (to $O_i$ estimated to be 0.26 eV). Charge redistributions in $H_2$ and with the surrounding Si bonds suggest a possible alloying of oxygen states with the Si band near the u-bridge. However, if such an alloying might be effective at room temperature, why did it disappear at the lower temperature of 77 K?

It might be noticed that a mobile motion of the hydrogen molecules accelerates the conversion rate [59,86]. It occurs in more disordered silicon samples when the molecule escapes from its cage, or when the silicon sample was heated at room temperature or as measured on the ASW samples (even if the conversion rates and the measurements temperature ranges were completely different).

### 3.2. Porous Catalysts

Porous Catalysts such as Metal-Organic Frameworks and Polymers have proved to be very efficient catalysts for ortho-para conversion.

### 3.2.1. $H_2$ Adsorbed in Metal-Organic Frameworks

S. FitzGerald and his team produced, over the last 20 years, an important amount of IR measurements of hydrogen molecules either inserted in $C_{60}$ [93,94] or physisorbed in MOF structures [46,48,95–97]. MOF-74 has two main primary and secondary sites where hydrogen can be adsorbed and these appear as two main strong absorption bands. MOF-74 exhibits hexagonal symmetry and takes on a honeycomb structure. Features with large redshifts appear at the lowest concentrations and are assigned to $H_2$ adsorbed on preferential sites, i.e., those with the largest site-specific adsorption enthalpy. There are three spectral regions of interest where $H_2$ vibrational and rovibrational transitions. In MOF74, three bands are found at 4088, 4092, and 4096 $cm^{-1}$.

Spectra-containing bands assigned to the purely vibrational Q transitions of adsorbed $H_2$ in MOF-74 are presented in Figure 6. As the material is loaded further, all three bands grow in intensity until the stoichiometry approaches 1 $H_2$ per Zn cation, after which there is a small redistribution of intensity represented in Figure 7 from the 4088 and 4092 $cm^{-1}$ bands to the 4096 $cm^{-1}$ band. This change in intensity was attributed to the ortho to para conversion process. Additional information is provided by the S lines also represented in Figure 7. The most notable point is that above a concentration of 1 $H_2$ per Zn there is no additional gain in intensity by these bands, indicating that the crystallographic adsorption site with which they are associated is saturated. Liu et al. found the complete occupation of one adsorption site at a concentration of 1 $D_2$ per Zn while other possible sites remained vacant [49]. The position of this site is directly above the base of the square pyramidal $ZnO_5$ coordination environment, the most reasonable site for preferred adsorption because it is the position of maximal electrostatic interaction. Crystallographically, it can accommodate one adsorbate molecule per Zn; these consistencies with different observations lead S. FitzGerald to assign the three bands at 4088, 4092, and 4096 $cm^{-1}$ to $H_2$ adsorbing on this primary site. The second series of bands appear in the frequency range 4128–4137 $cm^{-1}$ as the material is loaded with more hydrogen. They first appear at a loading of 0.85 $H_2$ per Zn, close to where the primary site saturates, and continue to increase in intensity with the addition of more $H_2$. That these bands also exhibit a smaller redshift from the gas-phase frequencies attests to their origin as that of $H_2$ on secondary adsorption sites. With reference to the neutron diffraction measurement, the secondary adsorption site is positioned above an edge of the $ZnO_5$ square pyramid, with the closest oxygen 3.4 Å away, and another is above an edge of the benzenoid ring, with two carbons 3.1 Å away. The occupation of the secondary sites also has a significant effect on the bands arising from molecules adsorbed

at the primary site and points to positive reinforcement of this site's adsorption enthalpy by adsorbate–adsorbate interactions. According to the crystallographic measurements, the inter-site distance between neighboring symmetry equivalents of the primary site is 5.3 Å, thus they are quite isolated. By comparison, the secondary sites are 3.2 Å away the distances changing somewhat with loading.

In each band, both ortho and para components are resolved, with the para component at an expectedly higher frequency. It was observed that conversion occurred with similar rates for molecules adsorbed either on metal or oxygen sites (of the order of the minute). It was also noticed that the metallic rates (above Zn ions) were enhanced by the presence of neighbor molecules adsorbed over the oxygen ions. A mysterious pattern was that important conversion had already occurred before the first minute necessary to register it, already also noticed by Fukutani on Ag samples!

Another promising investigation consisted in substituting the Zn ions in the MOF structure with several metal di-cations: Mg, Mn, Fe, Co, Ni, offering the opportunity to correlate almost linearly the infrared Q(0) line-shifts to adsorption enthalpies at zero surface coverage [96,97]. This isosteric enthalpy increasing from $-8.5$ to $-13$ kJ/mol, the authors statement is that microporous adsorbents with adsorption enthalpies surpassing $-15$ kJ/mol may be within reach. This would allow important progress in hydrogen storage.

The strongest center-of-mass translational mode centered at 4215 cm$^{-1}$, has a concentration dependence associated with the primary site and as shown for other systems, the broad bands arise from transitions in which the adsorbed $H_2$ increases its center-of-mass translational motion, commented by S. FitzGerald as "rattling back and forth within the adsorption site by one quantum".

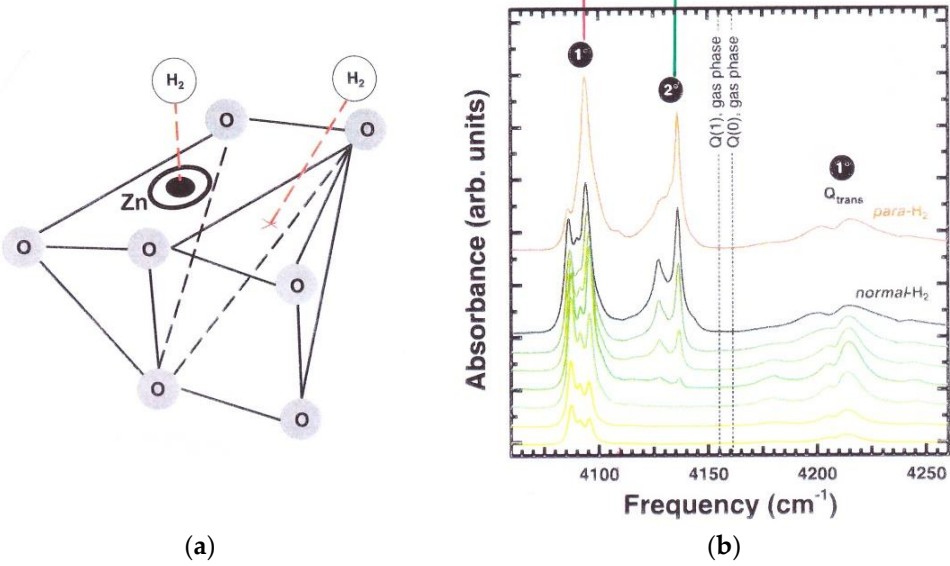

(**a**)           (**b**)

**Figure 6.** Hydrogen Adsorption and IR absorption Spectra in Molecular Organic Frameworks. Hydrogen Molecules adsorbed on the the sites: Primary (1) above the Zn ion and Secondary (2) above an edge of the $ZnO_5$ square pyramid: (**a**) Geometry (**b**) Infrared Adsorption bands. Reproduced with permission from [48] as follows: FitzGerald, S.A., et al., Phys. Rev. B, 2010, B81, 104305, Copyright (2010) by the American Physical Society.

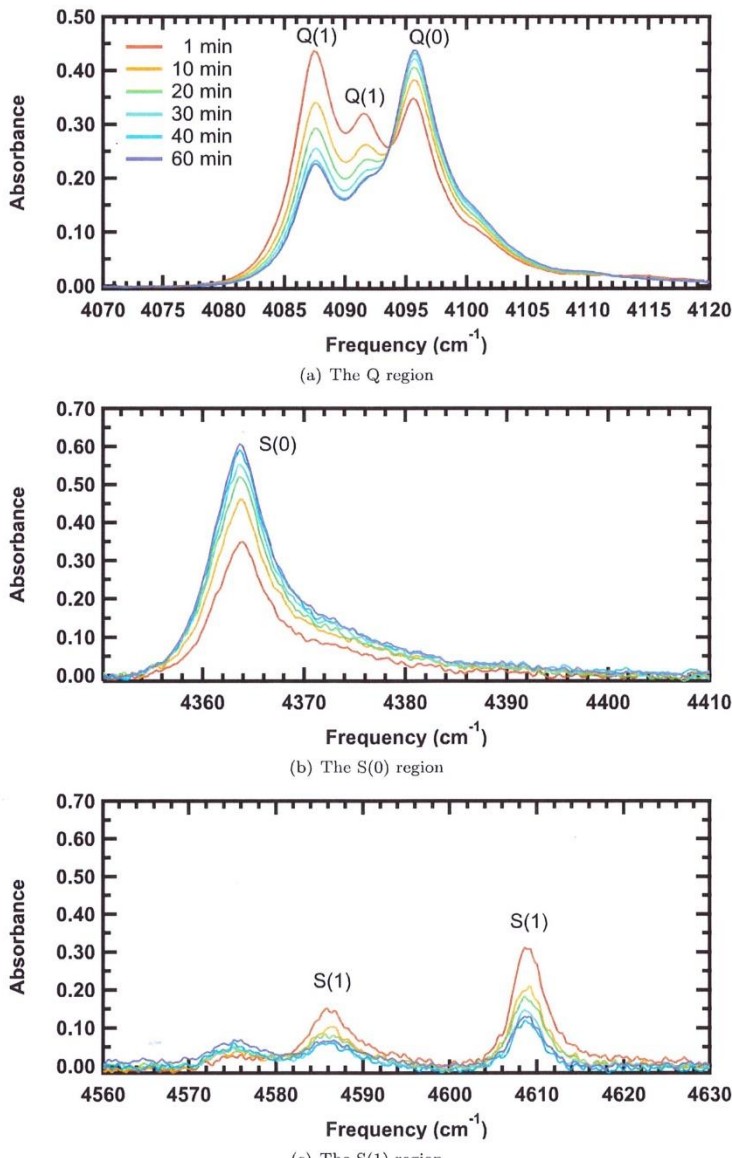

**Figure 7.** Ortho and Para Hydrogen IR Absorption Lines Time Evolution in Molecular Organic Frameworks. Ortho (red) and Para (blue) lines of Hydrogen adsorbed on MOF 74 sites (1). Ortho-para evolution in real time: The Q(1) and S(1) peaks decrease with time, while the Q(0) and S(0) peaks increase (The legend gives time after loading $H_2$ into the sample chamber). Reproduced with permission from [95], Burkholder, B., "Catalysis of Conversion Between the Spin Isomers of $H_2$ by MOF-74", Oberlin College, 2009.

### 3.2.2. $H_2$ Diluted into Polymers

Over the last 10 years, new STM (Scanning Tunneling) technics [98–101] or a combination of technics improved the observation of the rotational patterns of molecules inserted in nanocages. In particular, an interconversion of ortho (o)-para (p) nuclear-spin isomers for hydrogen molecule $H_2$ adsorbed in a nanosized space in porous coordination polymers (PCPs) was observed and measured by the temperature dependence of the Raman spectra [102]. Such report is also very original in the establishment of multiple identifications and correlations in between the progressive loading of the molecules on various sites, Raman spectra, ortho-para patterns and charge density measurements by X-ray diffraction. First, as no overlap of the charge densities was observed between the adsorbed $H_2$ molecules and framework, it was deduced that $H_2$ molecules are physically trapped in the pores (not necessarily at absolute equilibrium positions). These pores are considered as

electric nanocages. The following summarizes the relationships established between the measurements and the logical deductions described in that beautiful report.

Fast o-p conversions of the order of one or a few minutes have been observed on $H_2$ physisorbed over PCP [102], as on amorphous solid water [50]. Three $H_2$ sites were found, called sites -I, -II and -III. The peak at 65 K was found at the center of the pore on the (001) plane.

Eight charge density peaks between the pz rings, along a- and b-axes, were observed at 65 K, but disappeared at 35 K, the central peak at pore extended to diagonal. The charge densities of $H_2$ at 65 K correspond to the mixture of sites-I and -II. The $H_2$ densities at 35 K are interpreted as the mixture of sites-I and -III. The nearest inter-site distance of the sites-I-II is 3.68 Å and between sites I-III 2.060 Å. The $H_2$-$H_2$ distance in the pore has been estimated 3.0 Å to be the shortest distance between two $H_2$ molecules because of their repulsion. Thus, the sites-I and -II can coexist in a pore and the arrangement was achieved at 65 K with the adsorbed $H_2$ amount of 2.7 $H_2$ per pore, consistent with a maximum amount of 3 in this arrangement.

Charge densities at site-I were larger than those at site-II. A number of 2.7 $H_2$ per pore was achieved by combining fully occupied site-I and partially occupied site-II. The hydrogen molecules at sites-I and -II moved to site-III from 65 to 35 K. The main site of $H_2$ at 35 K is site-III. Site exchanges occurred only with cooling from 65 to 35 K. It is impossible to coexist sites-I, -II and -III in one pore, since the inter-site distances are too short. In addition, site-II cannot coexist with site-III of an adjoining occupied pore because of the too-short $H_2$–$H_2$ distance (2.43 Å).

In many of the pores, $H_2$ occupied site-III. The maximum amount of $H_2$ per pore for site-III is four. In the rest of the pores, $H_2$ mainly occupied site-I. Approximately 75% of $H_2$ was located at site-III at 35 K as deduced from charge densities and adsorption isotherms. The temperature dependence of Raman spectra in the $H_2$ adsorbed state was analyzed in correlation with the nuclear-spin populations and the structural change of the $H_2$ arrangement observed by SR-PXRD. Changes were found in the intensity and peak position for the Raman bands with cooling from 77 to 20 K.

The electrostatic potential around site-I was almost spherical. Site-I is only one minimum of the electrostatic potential in the pore. There can be the highest interaction between the adsorbed $H_2$ and the host framework at site-I. Therefore, peak A was assigned to p-$H_2$ at site-I, and peak C was also assigned to o-$H_2$ at site-I.

Site-I was occupied by a mixture of o- and p-$H_2$ at 77 and 65 K. The ratio of o- to p-$H_2$ from Raman intensities at site-I is approximately 3:2 which is the midpoint between equilibrium ratios at 77 K (1:1) and at 300 K (3:1) of normal $H_2$. Relative peak intensities of A and C did not change from 77 to 65 K. This is also consistent with full occupation of site-I at 77 and 65 K. Peak B at 35 K was easily assigned to site-III from Raman spectra and the structure. Therefore, peak B at 77 and 65 K can be assigned to p-$H_2$ at site-II. Relative intensities of peak B increased from 77 to 65 K consistent with a partial occupation of site-II. In addition, peak B did not shift from 77 to 65 K. The shift of peak B was assigned to the migration of $H_2$ between site-II and site-III. The Raman peaks around 600 cm$^{-1}$ had almost disappeared at 35 and 20 K indicating that o-$H_2$ was converted to p-$H_2$. The intensity ratio between peak C and a sum of the peaks A and B from profile fittings was less than 7% at 35 K and 1.2% at 20 K. The result supports that most of the $H_2$ at sites-II and -III are p-$H_2$.

Two principal processes were proposed to induce the conversion from o-$H_2$ to p-$H_2$ in the described PCP: (i) trapping of $H_2$ at sites-II and -III, and (ii) site exchange from site-I and -II to site-III by cooling. There is a huge difference between electric field magnitudes and of the charge distribution at the sites between site-I and sites-II, -III and the authors underlined such correlation between the electric field and the catalytic hydrogen o-p conversion at the sites.

The structural differences between sites-I, -II and -III were investigated using the charge densities and electrostatic potentials experimentally determined. The receiving

electric fields of $H_2$ from the framework can be estimated to be 0 for site-I because of the inversion symmetry, $7.4 \times 10^{10}$ Vm$^{-1}$ and $7.6 \times 10^{10}$ Vm$^{-1}$, for sites-II and -III respectively.

### 3.3. $H_2$ in Solid Nano-Cages

Three main methods were used to measure the concentrations of the ortho and para varieties, and their interconversion for $H_2$ molecules encapsulated in fullerenes: (i) the IR absorption lines intensity and dynamics, described in a previous section, (ii) Neutron scattering experiments, as carried out at CERN and described by the group based in Nottingham, were performed for hydrogen in open-fullerene at cryogenic temperature, (iii) NMR methods are also able to measure the protons spin magnetization and their relaxation in $H_2@C_{60}$ nanocages [103–117]. Insulator cages are able to store para-hydrogen with long dormancy (the cage shielding protects and increases the p-$H_2$ conservation). The advancement of "molecular surgery" has succeeded for instance to trap the hydrogen molecules inside fullerene cages, and then encapsulate them in a NaY zeolite structure where each pore (diameter around 8 Å) is able to accommodate one $H_2@C_{60}$ [103].

Due to the large, spherical shape of the molecules, solid $C_{60}$ has large interstitial voids making it a good host for matrix isolation. These voids come in two varieties. The larger of the two (2.02 Å), the octahedral sites, have ideal size for studying the dynamics of $H_2$ molecules (1.2 Å) because the sites are large enough that a hydrogen molecule can be trapped, resulting in quantized translational motion, and can rotate nearly freely within the site. On the other hand, the sites are also small enough that each will contain only one hydrogen molecule thus eliminating $H_2$–$H_2$ interactions. Using infrared spectroscopy to study $H_2$ intercalated within a $C_{60}$ lattice gives insight into the nature of the $C_{60}$–$H_2$ interaction. $H_2$ is not infra-reactive under normal conditions and so the $H_2$ absorption peaks in the spectra are purely due to interaction with the $C_{60}$ host.

The « Columbia » research group of Pr. Turro developed a new and fruitful molecular surgery (i) by opening the buckyballs (ii) inserting the hydrogen at high temperature and pressure through the created holes (iii) closing the holes and regenerating $C_{60}$ at room temperature. They sampled the $H_2$ molecules encapsulated in $C_{60}$ cages: $H_2@C_{60}$, and measured their conversion and relaxation rates by NMR techniques [105,107,111]. From the temperature-dependent relaxation times analysis, it was found that $H_2$ reorients itself about an order of magnitude faster in solution than as a guest in $H_2@C_{60}$, as might be expected for a very small molecule tumbling in the "soft" solvent cage, compared with the closed confines of the "hard" walls of $C_{60}$. Thus, in a simplistic interpretation, the incarcerated $H_2$ "feels" the walls of the $C_{60}$ cage to a greater extent than in a solvent. Note that the reorientation of $H_2$ in $C_{60}$ is somewhat faster than angular momentum change at the same temperature. This is consistent with a picture of the hydrogen molecule as a rotating gyroscope whose direction of rotation is changed by collisions with the wall faster than changes of the rotation rate. Different methods for enriching $H_2@C_{60}$ in the para isomer have been developed: (i) Cooling a sample of the fullerene in liquid $O_2$ and rapidly boiling off the $O_2$ after low-temperature equilibrium is achieved. Dispersal of the fullerene in a zeolite matrix was necessary to promote complete contact with the liquid oxygen catalyst. The enriched trapped $H_2$ could then be rapidly extracted before a conversion could take place. (ii) Attaching a functional group to the fullerene cage that can be converted to a paramagnet capable of producing a magnetic field gradient across the trapped $H_2$ molecule. Following low-temperature equilibration, the catalytic group must be rapidly back converted to the non-magnetic form. This method has been termed « a magnetic switch ». (iii) Low-Temperature photoexcitation of a fullerene containing trapped $H_2$ to paramagnetic triplets. Rapid decay of the triplet and re-excitation of the endofullerene lead to equilibration of the sample at low temperature. This method has been termed « a photochemical on–off switch » for o-p conversion.

$H_2@C_{60}$ inserted inside a zeolite framework was immersed in an oxygen liquid bath (at 77 K) to be enriched in the para manifold, then the oxygen was evaporated and the sample brought back to room temperature. The para lifetime otherwise measured at

7.5 days at room temperature was found in $H_2@C_{60}$ reduced to 100 h and further reduced to about 30 min when the cages were immersed in liquid oxygen at 77 K.

The comparison of $H_2@Si$ and $H_2@C_{60}$ nanocages complements our understanding of the interactions between molecular hydrogen and curved nano-surfaces. Different models have fitted the positions and intensities of the IR absorption lines by solving the eigenstates of a complex vibrational system. The quantized vibrations, rotations, and translational motions couple one to another. However, we may notice that the far-infra-red $\Delta N = 1$ transitions, predicted by the models to occur in the ground vibrational state $v = 0$ of the $H_2@C_{60}$ complex, were not observed (at the difference of the fundamental ones $\Delta N = 0$).

The coupling with the electron system is never explicit but a dipole moment was estimated by fitting the 4255 $cm^{-1}$ para line and associated to an effective charge of 0.006 $e$, and a displacement of 0.36 Å. For comparison, the amplitude of the internuclear oscillations in $H_2@Si$ exceeds 0.05 Å (0.5 Å for the CM motion). Both H atoms carry a small positive charge (less than 0.1 $e$, but larger than inside the fullerenes) and some electron density is transferred to their Si neighbors. This weakens the H–H bond and lowers the stretch frequency by a considerable 550 $cm^{-1}$. The time dependence of the induced dipole moment is associated with the oscillating internuclear distance every few hundred femtoseconds, rather than with changes in the effective charges. The $H_2@Si$ cage, compared to $H_2@C_{60}$ one, adds two couplings: the molecular vibrational system with the Si phonons and a collective electron-share between the silicon and the molecules. Both increase the incoherence of the relative random motion.

Nuclear magnetic resonance measurements in $H_2@C_{60}$ nanocages strengthen our interpretation of the $H_2@Si$ conversion patterns. First, the spin-lattice relaxation time exhibits a temperature-dependent maximum around 260 K which was conducted by Turro et al. [110] to conclude that the relaxation mechanism at high temperature is due to an intramolecular spin-rotation interaction. The conversion time was about 185 h at 300 K [116], close to the one observed in silicon cages and only slightly reduced to 100 h in presence of the $O_2$ catalyst. Consequently, an important property of the fullerene cage is its magnetic shielding and thus insulator cages are able to store para-hydrogen with long dormancy. It also confirms the effectiveness of the fullerene cage as a « bottle » for storing hydrogen-enriched in one of the spin isomers.

*3.4. $H_2$ in Viscous Organic Solutions*

It appears valuable to compare the conversion processes induced by solids with those occurring inside liquids and particularly in viscous solutions, since it gives information on how the hydrogen molecule behaves and "thermalize" when enclosed in a surrounding cage [118–124]. The first important NMR measurements of $H_2$ conversion catalyzed by paramagnetic ions in deuterated solvents were performed in 2005. Matsumoto and Espenson measured the o-p rate constants by means of the proton NMR in deuterated solvents at 298 K [125]. Most of the conversion rate data were correlated to the magnetic moments of the metallic ions and to the reciprocal of the sixth power of the collision distance.

NMR spectroscopy has proven to be one of the most powerful methods for understanding the mechanisms of hydrogenation reactions; indeed, it is widely used in catalysis for the characterization of homogeneous and heterogeneous catalysts, reactants and products in different phases, reaction intermediates. However, one of the major disadvantages of conventional NMR is its low detection sensitivity, which results from weak nuclear spin polarization under typical thermal equilibrium conditions. In order to overcome this sensitivity problem, several hyperpolarization techniques have been developed, including parahydrogen-induced polarization (PHIP) which is a powerful technique for studying hydrogenation reactions in gas and liquid phases and the examination of hydrogenation catalysts, as well as the mechanisms of hydrogenation reactions, remain a highly important task.

The subject of "singlet and triplet relaxations" and their inter-relations must also be replaced in a wider context: conversion and relaxation rates were measured in fullerene

cages and organic solvents by NMR. These measurements were followed by those of the Canet's group in between 2007 and 2017 [122–124]., which were the subject of my recent interpretation [126]. The important observation of Canet's group was that the o-$H_2$ proton NMR signal appeared immediately as a negative peak and after a while returns gradually to its positive equilibrium positive value.

The phenomenon was observed by the introduction of para-$H_2$ in a solution of a deuterated organic solvent (acetone-d6), with added paramagnetic entities such as $Cr(acac)_3$. A similar phenomenon was also observed for $CuSO_4$-$5H_2O$ and even residual $O_2$ resulting from an incomplete degassing of the solution prior to the p-$H_2$ introduction. As expected, the value of the p-o conversion time was found to decrease with increasing copper ions concentration. At ambient temperature and with 100% enriched parahydrogen (p-$H_2$) dissolved in paramagnetic organic solvents, the o-p spin conversion was observed to occur in between 4 min and 1 h depending on the magnetic concentration of the solution. The experimental data were fitted according to exponential relaxations, with characteristic times denoted by these authors the para-hydrogen relaxation times. For the magnetic solutions investigated by Canet's group, the values of both relaxation times (corresponding to the transitions inside the ortho manifold or between the ortho and para ones) decrease as expected with increasing magnetic ions concentration. However, their ratio is not constant as was sometimes postulated. When the concentration is multiplied by 35 the conversion rate is multiplied by a factor ~43, whereas the relaxation rate only by a factor ~6. The relaxation time varies more slightly with the magnetic concentration, because of its essential intramolecular origin. I shall return briefly in Section 4.1.1.2. to the theoretical interpretation of these experiments, but at this stage, it is important to remark that the hydrogen molecules and the magnetic impurities recurrently stick together for a short time. The hydrogen molecule is then confined inside a nanocage whose size depends both on the solvent viscosity and the attraction of the metallic magnetic impurity. Consequently, the relative values of the relaxation times are related to the ratio of the long- and short-range interactions, as well as to the size of the nanocage and the time spent by the molecule within the cage.

To conclude that chapter, I remark that most of the recent studies on hydrogen encapsulated inside nanocages describe the molecular motion from a mechanistic point of view. The systems are described by a coupling between internal and external vibrations with quantified translations inside the cages bounded by hard walls. These limit conditions should be explicated more extensively. Despite the fact that the experimental procedures being performed by electromagnetic measurements, the theoretical analyses do not take into account the electronic structure of the molecule–cage partners and their electronic interactions are scarcely taken into account. Even the molecule-induced dipole moments are rarely estimated, which remains particularly questionable in the case of the molecules interacting with ionic solids.

## 4. Qualitative Analyses and Theoretical Advances

In that chapter, I summarize the new "conversion" concepts that have emerged. First, the conversion process is an electromagnetic one and the hyperfine interactions involve the magnetic as well as the electric ones. Second, the catalyzed hydrogen conversion on a solid is a collective phenomenon that involves the catalyst electronic structure. Third, the ortho-para conversion of hydrogen molecules occurs when the initial preparation is not at equilibrium with the surrounding thermal bath, and therefore has the character of irreversible thermodynamical processes. In the intermediate situation of nanocages, the degree of molecular freedom of the molecules and their thermal accommodation depends on the size of the cage and on the electron « hardness » of the walls. In viscous solutions, long and short-range interactions compete and complement each other, in a proportion function of the viscosity of the medium.

The ratio of "localized/collective phenomena" at the surface of a catalyst is particular in the new "Meta-sorption" regime, an intermediate region between physical and chemical

adsorptions characterized by high catalyst–molecule adsorption heats (isosteric enthalpy from −8 to −15 kJ/mol), even in absence of potential minima. The molecule might be encaged although not linked, mixing their electronic antibonding excited states with the catalyst ones. Although not occupied, they allow transient fluxes and transfers of energies and momenta. In the case of adsorption, the molecule bounces on hard walls and in very short times is submitted to intense electronic repulsions that involve the solid excitonic spectrum. In some new way, the electronic and thermal degrees of freedom are linked dynamically and that link is observed in the hyperfine conversion rate measurements.

In Section 1 of that chapter, I stress the electromagnetic character of the conversion processes and in particular the antibonding alloying in between the molecular excited states and the solid conduction ones. In Section 2, I underline the collective character of the energy and momenta transfers involved in the catalytic reaction.

### 4.1. Electromagnetic Hyperfine Catalysis

Hyperfine interactions measure the nuclear dynamics driven by the electronic ones. From the point of view of a qualitative theoretical physicist, I am attached to relate formalism and concepts and have searched a language and an algebra to describe the conversion process and their rate estimates [127–134]. These processes were classified and each one was described by a channel. These channels involve catalytic steps. Their number: 1, 2 or 3 step-mechanisms are related to the number of links in the reaction path or to the number of interactions as well as to the order of the time-dependent perturbation theory. One-step channels for instance are illustrated by the **D** (dipolar magnetic) and **Y** (hyperfine contact) mechanisms. Two-step channels are illustrated by the **CY, XY, UY, LY** . . . mechanisms, where **C** represents the Coulomb repulsion (with electron exchange **X** or charge transfer **U**); **L** represents the electron orbital–molecular rotation interaction. In between the paired interactions, a projector is introduced to span the electron-excited spectrum. The tensorial coupling of the interactions is treated within the 3 dimensional–rotational group algebra. For the third-step channel **CSY,** a non-diagonal electron spin–orbit interaction **S** is added, so that the catalytic path brings the system back to its fundamental ground state. Therefore, the more local mechanisms of 1 and 3 steps lead to a nuclear transition whereas the 2-step ones lead to electronic energy dissipation through the catalyst [24,35–40,127–134].

#### 4.1.1. Paramagnetic Conversion
#### 4.1.1.1. Magnetic Catalysis on Solid Surfaces

Magnetic Conversion studied since 1933 has denoted the hydrogen conversion induced by a first-order magnetic interaction between a hydrogen Molecule and the magnetic moment of usually a localized ionic impurity. The dipolar processes gave a simple interpretation of the conversion measurements on a large variety of magnetic substrates all along the last century.

In the seventies [16–20] and later in the eighties [23–26], I extended the quantum formulation of the conversion process, applied it to interpret experimental measurements of the conversion speed observed for Chromia, Nickel and various 3d magnetic ions. In particular, I interpreted the magnetic field effects observed by P.W. Selwood on the basis of the electron spin–orbit fine structure [27]. Experimental research on $H_2$ catalysis and relaxation remained seldom up to the new century and limited. On the theoretical side, decisive progress was realized when I started a new collaboration with a few Japanese teams.

In 1993, the orbital degrees of freedom of the surface electrons and their spatial extensions were studied further in collaboration with Osaka University and K. Makoshi. Our general expressions were illustrated by a simple model and discussed in terms of a limited number of parameters: the surface-molecule distance d, the effective metal nuclear charge Z and different orbital geometries were compared [135–141]. Owing to the catalyst electron localization and molecule-surface distance d, the magnetic catalytic rates were compared to the Wigner law: conversion rate $\sim d^{-8}$. In particular, the o-p conversion rate was found to be enhanced by the presence of a metal dangling bond perpendicular to the

surface [135–137]. Figure 8 represents and compares the one-electron hydrogen conversion rate dipolar (**Ds**), contact (**Y**) and Overlap-Contact (**OY**).

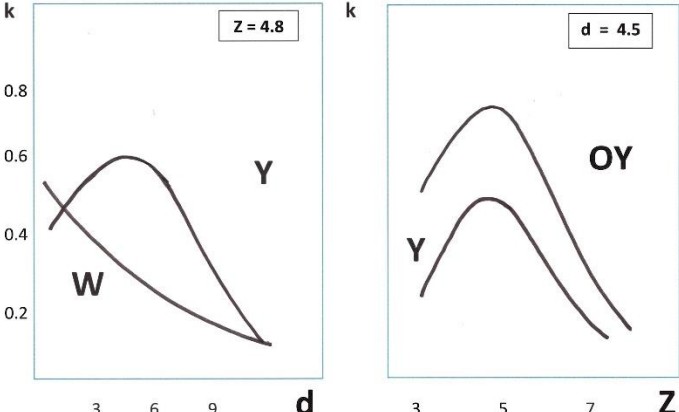

**Figure 8.** Dipolar, Contact and Overlap-Contact Conversion Rates of $H_2@Cr^{3+}(O^{2-})_5$. A Hydrogen Molecule above a $3d^3$ magnetic $Cr^{3+}$ ion at the center of the pyramidal $(O^{2-})_5$. The relative effect of the magnetic ion spatial extension (represented by an effective nuclear charge Z) on the conversion rates(K in $s^{-1}$), is illustrated by a comparison between the infinitely localized **W** dipolar model and the diffusive Contact Interactions **Y** and **OY**, and is represented by functions of the "molecule-magnetic ion" distance d and the electron orbital spatial extensions described by a Slater basis.

A little earlier, in 1986 S. Sugano spent a full year in my laboratory of Surface Magnetism, and we considered the spatial extension of the catalyst electrons. For the first time, the hydrogen conversion was not assumed to be induced by magnetic point dipoles. First, the electron molecular and surface orbitals were introduced, the dipolar and contact interactions compared, and the overlap–contact interaction was considered. Since the molecular and surface electrons overlap slightly, a surface electron can contact the adsorbed protons through a molecular orbital. In the coupled electron basis, the molecular and surface orbitals are not orthogonal. The hydrogen conversion could be discussed first in terms of one-electron matrix elements: those of the dipolar (**D**), contact (**Y**) and overlap–contact (**OY**) interactions. However, after realizing that the molecule–metal electron overlap could strengthen the contact process considerably, we succeeded in major progress in introducing the two-step mechanisms (**XY**) [132–134].

In a first step, the molecule–solid electron repulsion excites the molecule in a virtual state and evacuates the $H_2$ rotation momenta. The singlet ground state of the coupled system is disintegrated, by the Coulomb interactions among electrons, into two excited triplets. In the second step, the magnetic hyperfine contact links the "s" excited electrons with the nuclei. Here a new concept aroused in the conversion theory: the hyperfine interaction cannot be reduced to a purely nuclear transition but involves also an electronic one.

Nuclear perturbations are so weak that even a very small electronic effect might compete with the nuclear disturbance. We denoted that second-order process: the **XY** exchange-contact conversion. For the **XY** channel, the molecule–surface electron system is coupled, since these electrons cannot be distinguished and might exchange one another.

This is a phenomenon similar, but opposite, to the triplet–triplet annihilations involved in fluorescent emissions occurring in radical pairs. The electromagnetic radiation is replaced by the rapid variation of the surface inhomogeneous electrostatic field felt by the vibrating molecule. The **XY** channel was suggested to interpret the conversion rates of hydrogen adsorbed in front of a chromium impurity inserted in an alumina surface [132–134].

4.1.1.2. Magnetic Catalysis in Solvent Solutions

The method of magnetic resonant pulses has proved to be efficient in detecting the magnetic transients of the hydrogen conversion, and the observed para → ortho conversion

induced by the introduction of non-equilibrium hydrogen mixtures in a liquid solution implies new magnetic flows that were observed by Canet's group [122–124].

When para-hydrogen molecules prepared at very low temperature are diluted in a paramagnetic liquid solution at room temperature, an irreversible angular momentum flows towards the hydrogen protons and a resulting nuclear polarization can be observed during a delayed time characterizing the solvent mixture. I introduced the concept of a non-equilibrium "ortho drift" to interpret the observed transient nuclear magnetization directed along the electron spins and thus opposite to the applied magnetic field.

It appeared convenient to define a microscopic conversion rate, representative of a process where the partners are in relative contact, as if the hydrogen was confined within a nanocage, and to distinguish it from the NMR measurement of the conversion process delayed by the solvent. The important fact is that the nuclear memory is conserved between two scatterings and that the nuclear magnetization follows the ortho population's return to equilibrium only in the long term.

Therefore, the negative peak of the o-$H_2$ NMR signal provided a unique opportunity to study and measure the time evolution of the $H_2$ singlet and triplet states and their two time scales, one of a few seconds and the other general of about one hour. By comparing these measurements with the theoretical analysis, it was possible to obtain information on the time spent by each hydrogen molecule inside the electron cage surrounding each impurity [126].

### 4.1.2. Metallic Physical Conversion

All the existing models, applied to metal oxides induced conversion before 1990, relied on catalyst magnetic moments, usually incomplete shells of transition metal ions: 3d or 4f ones inserted on a substrate. These processes were always assumed inefficient for diamagnetic surfaces, since these substrates have full and closed shells and no magnetic moments. Strangely hydrogen conversion was observed to occur on noble metals by EELS (electron energy loss spectroscopy) measurements in 1982 [32,33].

First P. Avouris at IBM labs reported that the ortho molecules deposited on a surface of silver (111) had disappeared and probably converted in their para form, before he could register the transformation: a time of about 10 min was necessary to run the apparatus [32]. Shortly later, S. Anderson in Chalmers observed a slower effect on Cu (100) [33]. A similar effect was reported later on graphite [34]. In all cases, the rates were too fast to be registered. These experiments have established the primary observation of conversion on (i) non-magnetic catalysts (ii) metals without chemisorption (iii) single crystals covered by freely rotating hydrogen molecules.

I spent the sabbatical year of 1990 in the Research Center for Extreme Materials of Osaka University to understand why and how the hydrogen had been converted on the noble metal surfaces at the rather low temperature of 10–15 K, and why so fast on Ag(111). Most of the physicists did not believe that conversion had occurred on these surfaces and attributed the observation to the presence of remained impurities.

My first challenge was to understand how a surface–molecule interaction could occur on a non-magnetic substrate, which I concluded by: "the only possibility is that the final reaction state had been magnetic". In other words, the selection rule $\triangle S = S_f - S_i = 0$ must be replaced by $\triangle S = 1$. The angular momentum conservation prescribes a transfer of rotational molecular momenta to the spin-orbital ones of the metal electrons. The interpretation of the IBM experiments was consequently led to explain how the metallic surface had emitted magnetic quasi-particles during the conversion of the adsorbed molecules.

The second interpretation step consisted of discovering the nature of such an emission. At 10 K a metal conduction electron might jump from an occupied state to an empty one, if located at above 14.7 meV (the o-p transition energy). That new excited electron couples to the hole left behind and creates an electron-hole pair that travels through the bulk. Such a pair can accommodate a magnetic quantum when the electron and the hole "parallels" their spins, that is form a triplet pair of spin 1.

Another step was necessary to understand the observed difference of about one to two orders of magnitude, between the Ag and Cu conversion times. Ag and Cu have similar band structures, but the surface orientations of the samples might have introduced important differences. The silver conversion rate had been measured over a (111) surface whereas the copper one was on a (100) one. Now the (111) surface state is well known to be active in a few surface reactions, due to an important density of states and a spatial tail less decreasing outside the surface than the bulk electron states. For sure the observed difference of conversion rates was not related to the bulk conduction bands but attributed to the relative efficiency of the (111) surface state.

The next interpretation step consisted of the elucidation of the magnetic mechanism. The first estimation of the dipolar interaction between the electron and proton spins led to a conversion time of the order of a month! Thereafter the consideration of the contact interaction reduced that time to a few days and when adding the electron molecular–surface overlap reduced further the time to 14 h. Clearly, the overlap was efficient. In the physisorption state, the molecular protons are too far away from the metal electrons. However, when the electron clouds overlap, a metal electron might be introduced in the molecular configuration. That non-orthogonality conducted to consider carefully the Coulomb repulsion among the surface complex. The best candidate for strong repulsion was the 1s molecular spin-orbital. Apart carrying an electron spin, it has important amplitude at the molecule protons.

The last interpretation step reduced to compare the various transients that might be virtually excited: electron exchange or charge transfers. In both cases the $H_2$ conversion is conducted in two steps: one electric by excitation of the molecular antibonding 1 s state, the other magnetic by the hyperfine contact. These two interactions break the two molecular selection rules: the electron interaction transfers the orbital rotational momenta and the magnetic one the nuclear spin ones.

The molecule–surface electron exchange was denoted **X**, the charge transfer **U** and the hyperfine contact **Y**; the related conversion processes were correspondingly denoted **XY** and **UY** [5,6,35–40]. The charge transfer process **UY** was found slightly faster than the electron exchange **XY**, estimated in between 1 and 10 min for (111) noble metal surfaces, and in between 20 min and 1 h for the bulk and (100) surface states. It was also predicted that a radiative excitation of the molecular surface layer would accelerate the conversion process.

The image surface states might also contribute if excited [40]. In 2003, 23 years later than the IBM observations (and 13 years later than my interpretation) the Japanese team of the Tokyo ISS team conducted by K. Fukutani, succeeded in measuring a conversion time of 13 min by REMPI on a polycrystalline silver sample at 15 K, and observed a slight conversion acceleration by applying a laser beam [41,42].

The Ortho-para Ratio measured as a function of the hydrogen molecules' residence time on the silver surface, by K. Fukutani and al. [41], is represented in Figure 9. Altogether, the conversion process is a "non-dissociative" metallic catalysis of hydrogen that should be distinguished from the usual chemical metallic catalysis of hydrogen which proceeds by dissociation.

### 4.1.3. Conversion in Dielectric Insulators

Many experimental observations of the hydrogen nuclear spin isomers have been reported in the past decades, but seldom theoretical ones [142,143]. Apart from the metallic conversion ones, new and numerous conversion effects were observed on amorphous or crystalline insulators, in semi-conductors, polymers and viscous organic liquids. The non-magnetic conversion mystery had started off again [127–131,144]. That section will deal with the fast conversion rates that had been measured in MOF and ASW samples at low and very low temperatures, with characteristic times of the order of one or a few minutes, as related in Chapter 3. These catalysts are non-magnetic and insulating. Consequently, the adaptation of the metallic process faced the presence of a gap of a few eV to cross, in

order to excite a surface electron to a molecular configuration. I was thus conducted to incorporate a larger number of electron states and interactions [127–131].

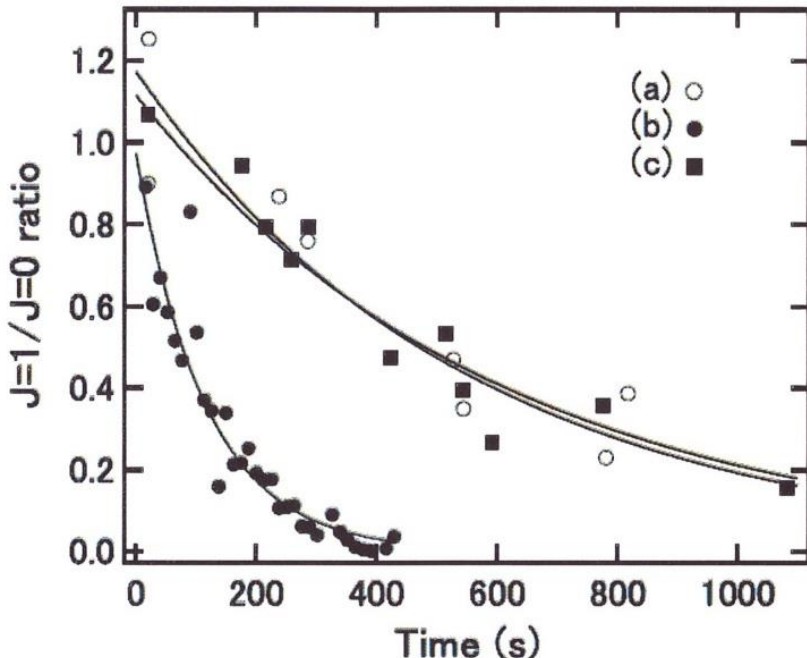

**Figure 9.** Hydrogen Ortho—Para Ratio Time Evolution on Silver (111) Noble Metal Surface. The o/p ratio of $H_2$ on an Ag surface at 7 K as a function of time probed by REMPI/PSD (**a**) Probed every 250 s, (**b**) under laser irradiation at 193 nm with a fluence of 120 J/cm2/pulse, (**c**) under laser irradiation at 532 nm with a fluence of 120 J/cm2/pulse. (alternate measurements of the ortho and para intensities). Reproduced with permission from [42] as follows: Niki, K., et al. Phys. Rev. 2008, B77, 201404, Copyright (2008) by the American Physical Society.

Two channels were investigated. In the first of the two steps, the **CY** process is recognized as the leading one for the hydrogen conversion on non-magnetic catalysts, although it leads to a final excited surface state. The created exciton propagates through the bulk until being annihilated by the solid phonons. In other words, the hyperfine conversion is driven by the high-frequency electron fluctuations. Such an **XY** channel that might operate on a diamagnetic catalyst is illustrated in Figure 10. The hyperfine molecular "flip-flop" between the electrons and the nuclei spins is similar to the one suggested for the magnetic catalysts.

In the second possible channel, a 3-step **SCY** process operates at the nuclear frequency. The initial ortho and final para states both belong to the electron ground state of the surface complex, but the reaction path crosses two intermediate virtual states. Such a process is constructed in bringing together the previous two Coulomb-Contact **CY** steps with the **S** spin–orbit interaction that links the electron excited in the conduction band to the hole left in the valence band.

I denoted "electronic" conversion the path involving emission of excitons that propagate and disintegrate in the bulk and denoted "nuclear", the path where the excited electron states are transients of a loop whereas the electron system returns to its fundamental ground state.

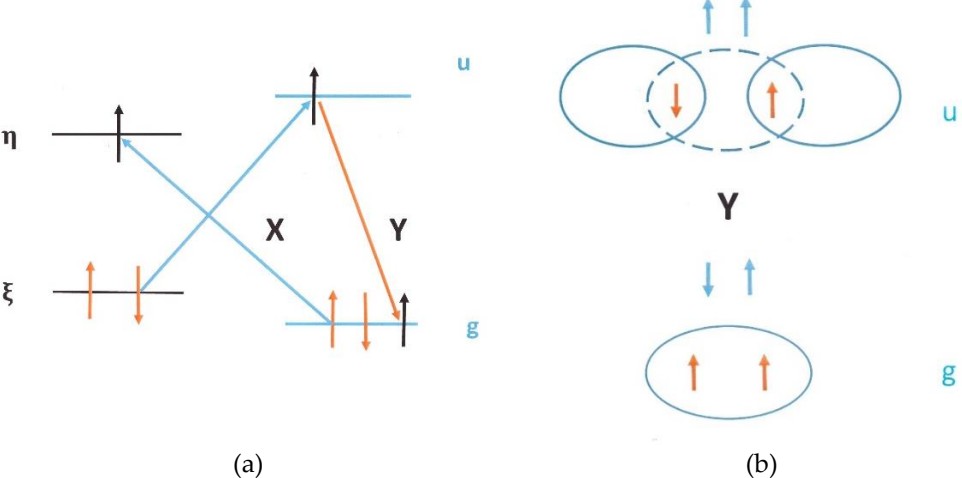

**Figure 10.** (**a**) **XY** Conversion Process: Electron Exchange **X** and Hyperfine Contact **Y** of Molecular Hydrogen interacting with a Diamagnetic Catalyst. The molecule–solid electron exchange C = **X** excites simultaneously the molecule and the solid antibonding states (a two-electron excitation). In such an intermediate state of that singlet channel of mutual neutral states, the bonding (and antibonding) electrons have parallel spins. (**b**) In the next step of mutual electron and nuclear spin "flip-flop", the molecular electrons transfer their spin momenta by hyperfine contact **Y** to the nuclei (the molecule rotates differently although not represented), converting the molecule from its initial para state to an ortho one. On the solid side, an exciton is emitted and propagates until mutual annihilation.

The **S** spin–orbit interaction had already been considered by Sugimoto and Fukutani, but as occurring within the molecular space. In 2011, when they observed the o-p conversion of hydrogen adsorbed on amorphous ice (ASW) around 10 K by the "REMPI" method (Resonance Enhanced Multi-Photon Ionization [52]). They attributed the fast o-p conversion that occurred in about 6–12 min [50] to the ASW giant surface electric fields. I denoted their suggested path: (**SOPY**), where **SO** is the spin–orbit coupling inside the molecular space enhanced by the centrifugal electric force that accelerates the electrons rotation around the protons, **P** the Stark coupling of the electric molecular dipole with the surface electric field and **Y** the electron-proton hyperfine contact. However, that path represented by the conjunction of the following perturbations **PYP²SP²** faces a few drawbacks. First, it necessitates the consideration of seven orders of perturbation by repeated action of the Stark action **P**, and would necessarily involve a summation over the full electron molecular spectrum. Secondly, the treatment of the molecule–surface interaction by a Stark coupling is particularly questionable in the case of strongly inhomogeneous surface electric fields. Therefore, after having studied in detail the **SOPY** conversion path [128], I attributed the fast conversion rates registered on ASW surfaces to the conjunction of only three couplings resulting in a third-order perturbation [131]. The corresponding **SCY** conversion path is represented in Figure 11 as an electron closed loop.

The **SCY** mechanism was studied in two directions: (i) in constructing the conversion quantum operators and (ii) analyzing the energy and momentum transfers. These time-dependent transfers, represented by the spectral density $J_\omega(\tau)$, will be discussed in the next section, devoted to the collective phenomena operating inside the catalyst. As considering the quantum part estimation of the conversion rate, the third-order time-dependent perturbation theory, applied to the "$H_2$-$H_2O$" electro-nuclear system, conducts to the conversion rate: $\mathcal{P}_{o\to p} = \mathbf{R}\, J_\omega(\tau)$, taking into account the nuclear summation of the squared matrix elements: $\mathbf{R} = h^{-2} \sum_{i,j} \left| p \left| \mathbf{O} \left| o_{i,j} \right. \right. \right|^2$, between the fundamental para state p and the ortho ones $o_{i,j}$.

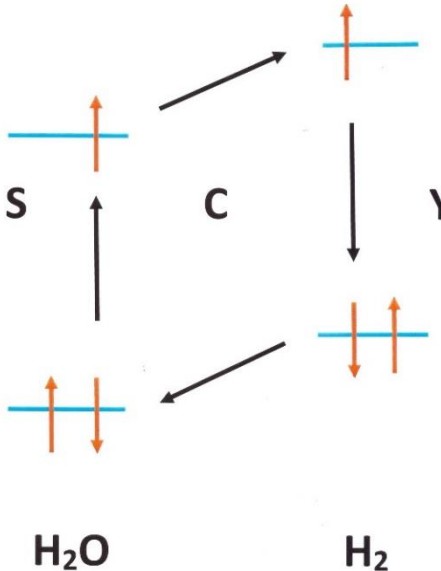

**Figure 11.** **SCY** Conversion Channel: Spin Orbit **S**(Λ)—Coulomb Repulsion **C**—and Hyperfine Contact **Y**. **SCY** Conversion Process for Molecular Hydrogen interacting with Amorphous Solid Water. In that 3-step **SCY** channel, a first one electron spin–orbit interaction **S** excites the surface water molecule in a 3 s orbital perpendicular to the surface. The electron repulsion with the hydrogen antibonding σ*$_u$ orbital induces then a simultaneous exchange of antibonding and bonding electrons. The last contact step relaxes the hydrogen molecule to its ground state, changing the initial ortho or para variety. Only the nuclear o-p energy has been exchanged.

The effective perturbation operator writes **O** = **SGCGY**, where the spin–orbit **S**. and the hyperfine **Y** admixtures connect the non-magnetic ground state to excited magnetic ones, whereas the electron repulsion **C** interchanges electron orbital motion and molecular rotation. G represents the Green projector summation over the whole electron « Molecule—ASW » system spectra. Qualitatively, the order of magnitude of the conversion rate is obtained from:

$$\mathcal{P}_{o \to p} \approx h^{-2} \left\{ \frac{\Lambda\,CY}{\varepsilon_{gu}\varepsilon_{ab}} \right\}^2 J_\omega(\tau)$$

with the figures: C $\approx$ 1 eV, Λ $\approx$ 3 meV ($\approx$ 20 cm$^{-1}$), Y $\approx$ 1.6 $\times 10^{-6}$ eV, $\varepsilon_{gu}\varepsilon_{ab} \approx$ 100 eV$^2$, (g(u) and a(b) represent respectively the molecule and solid fundamental bonding states (excited antibonding) and $\varepsilon_{gu}$, $\varepsilon_{ab}$ the associated transition energies). With $h\omega$ = 14.7 meV; $\tau \approx 3 \times 10^{-13}$ s at T = 14 K, the conversion rate is estimated as $\mathcal{P}_{o \to p} \approx 1.59 \times 10^{-3}$ s$^{-1}$ in remarkable agreement with the experimental value of 1.7 $\times 10^{-3}$ s$^{-1}$.

The interpretation of the conversion patterns of molecular hydrogen adsorbed on amorphous solid water confirms the **EM** electromagnetic nature of the process. The **E**lectric repulsion mixes bonding and antibonding electrons transferring rotation towards orbital momenta, whereas the **M**agnetic spin–orbit coupling and the hyperfine contact introduce some electron spin-orbital excitation in the H$_2$-ASW ground state transferring spin momenta from the virtually excited electron to the nuclear spin momenta. Qualitatively the charge-transfer channels are certainly enhanced by very-high surface electrostatic fields when they help the electrons getting closer and thus increase their repulsions.

In 2013, I enlarged the understanding of the processes at work to break the Pauli symmetry requirements. In some sense, the alumina-chromia impurities systems [14] and the amorphous ASW cases were simpler than the MOF or Silicon systems because the catalytic interaction is more localized at the surface. For regular structures, it appeared necessary to involve the full solid band structures [129,130].

The extension of the electron basis to charge-transfer states and "continui" of band states are important to focus on the broadening of the antibonding molecular excited state

by the solid conduction band since they provide efficient tunneling paths for the hydrogen conversion. The suggested virtual electron exchanges between the molecule and the solid gave a qualitative interpretation of the measures on ionic insulators, but the charge transfer processes also certainly contribute to the conversion rate. Quantitatively, the relative strength of the possible channels **XY**, **UY**, **SCY** remains an open question particularly in the Zn-MOF systems [93–97] in the pores with accessible $Zn^+$ and $O^-$ ions. On such samples, the conversion rate was fast, and almost as fast on the metallic and oxygen ions. Moreover, it was also noted an interaction between the hydrogen molecules adsorbed on adjacent sites which should be investigated further.

In the case of hydrogen physisorption on MOF complexes, a recent computational study has shown that the charge–transfer (CT) interaction to the antibonding molecular orbital (u = $\sigma_u(1s)$) is present in all studied MOF samples and contributes up to approximately −2 kJ/mol to the adsorption enthalpy [145]. Is it reasonable to consider such adsorption as physical?

I suggest the term "Meta-sorption" to point at the large region intermediate between physical and chemical adsorption. I shall return to that concept in the next Storage section since MOF is one of the possible strong adsorbents with practical use for FCEV transports. Here we are dealing with the interesting theoretical question, of the gradual transition from virtual charge transfers to real ones and their channels. Such a question has been well known in molecular spectroscopy and surface chemistry since 1940, when the importance of the "ungerade" excited states was recognized to describe the internal bond release. It appeared also recently in relation to the formation of the Molecular Clouds from interstellar grains and their dating.

It is widely accepted that $H_2$ molecules are created by H–H recombination reactions on dust. When one of the H atoms is in an electronic excited state the radiative association becomes allowed. When two H atoms encounter each other on the surface of dust, electronic excited states jump from one to another and third body effects operate. The charge–transfer **SUY** channel suggested to operate on the ASW surface is represented in Figure 11 and more generally on a diamagnetic insulating surface in Figure 12.

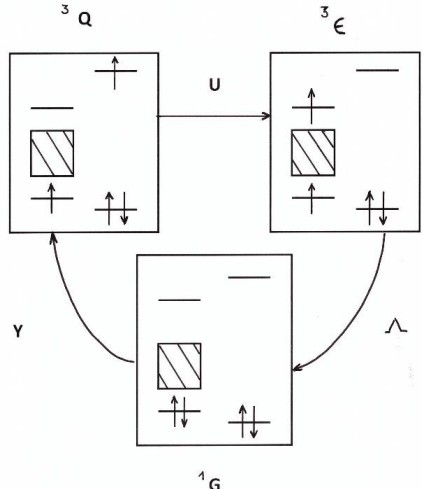

**Figure 12.** Charge Transfer **U** and **SUY** Conversion Process of Molecular Hydrogen interacting with a Diamagnetic Insulating Solid Catalyst. Triplet Channel: An example of a solid valence electron jumping on the molecular u state by hyperfine contact (a single-electron excitation). The shifted electron and its remaining pair in the solid build a magnetic electronic state that flips the nuclear spins. The electron repulsion brings back the shifted electron to the solid through the molecule–solid bridge (the bonding of the antibonding states) emitting (or not) an exciton. That tunneling is denoted as one **UY** channel. For appreciable SO coupling: Λ in the solid and sufficient scattering time, the solid relaxes before the molecule leaves leading to the **SUY** channel where only a nuclear energy is exchanged with the solid. Reproduced from [129].

The ortho/para ratio became a valuable signature of the molecule formation and a measure of the surface temperature. The conversion rate becomes a characteristic of how the thermal three systems share their temperature differences one with another: Molecule System (Quantum Structures) − Thermostat (dynamical surface motions: molecules + phonons) —Electronic System (molecular + conduction).

The new non-magnetic conversion patterns that appeared in the past twenty years in the new materials investigated are particularly related and sensitive to the surface electronic structures [144,145]. Are they related to the local electric field strength and/or to its inhomogeneity and/or to the local density of charges? These remain as important questions to investigate further.

### 4.1.4. Conversion in Nano-Cages

Let us now consider hydrogen molecules inside electron cages of different sizes and in particular diluted inside insulated cages such as fullerenes or semi-conductors. Most of the proposed mechanisms, ion–molecule electron exchanges **XY** and charge transfer **SCY**, **XY**, **UY** can be applied to the relaxations inside the nanocages. That theoretical framework unifies the different observations of o-p conversions on non-magnetic insulating catalysts.

The ortho-para conversion of hydrogen molecules oscillating inside tetrahedral cages of silicon compounds is an excellent system to study the interaction of the nuclear protons with the silicon electrons. At each collision against the cage hard walls, the electron repulsion changes the molecular rotation while projecting a valence electron in the antibonding molecular state dressed by a group of conduction ones. That « bridge » facilitates the hyperfine contact of the electrons with the protons. At room temperature, the angular momentum transfer is enhanced by electron fluctuations that overcome the silicon gap and accelerate the nuclear rates by more than one order of magnitude. The electron fluctuations seem to predominate at room temperature, as the shortest way to transfer the magnetic excitation, because of the weak gap and ionization energy of the Si structures. Static electron orbital momenta coupled to the nuclear vibrational system become more efficient at lower temperatures [130].

Platelets and $T_d$ sites are two extremes in terms of the trap size. Inside a solid, $H_2$ interacts differently than at the surface. It occurs in the bulk as a collective phenomenon in a double way: any excitation to the *CB* involves an orbital disturbance disseminated through the whole crystal, and thus interacts with all the « inside-molecules ». It is striking that we found average repulsion energies relatively close one to another for the molecules inside different silicon samples, and of the expected order of magnitude ($\approx$100 meV).

For silicon crystals (and oxygenated ones) the relatively weak orbital coupling strength of about 50–70 cm$^{-1}$ confirms the « electronuclear » channel efficiency at 300 K. For more disordered structures, where the $H_2$ molecules are trapped by multi-vacancies in ion-implanted silicon K complexes, stronger orbital couplings ($\geq$130 cm$^{-1}$) allow more efficient « nuclear » channels at 77 K. For the silicon platelets, characterized by fast conversion rates but lower thermal accelerations, both channels are probably effective at 77 K. Both channels **CY** and **CYS** are summarized and compared in Figure 13.

I agree completely with the important conclusion drawn by the « Lehigh » Stavola's group [47] that most of the $H_2$ conversion occurs while the molecule is dynamically off-center and essentially driven by their nuclear spin–rotation coupling, and thus I detailed the nature of that coupling [130]. For molecules inside the tetrahedral cages, numerical simulations of the molecule center of mass motion inside the silicon cages, performed in 2001, have indicated that the molecules follow an erratic motion at a "fantastic pace" [59,60,85]. The molecule is found to change abruptly of direction every few $10^{-13}$ s, while bouncing off the walls of their tetrahedral cage and exchanging energy with the host crystal. As the temperature is increased, the fluctuations of the potential energy of around 0.5 eV at 77 K also increase. The H–H bond weakens and oscillates by coupling the $H_2$ motion (center of mass and stretch mode) to the Si phonons. It seems that the incoherent motion in the cage does not reflect an accommodation with the solid temperature. The molecular motion is

more driven by the electric repulsions when they bounce the hard potential than by the phonon-vibration mismatch.

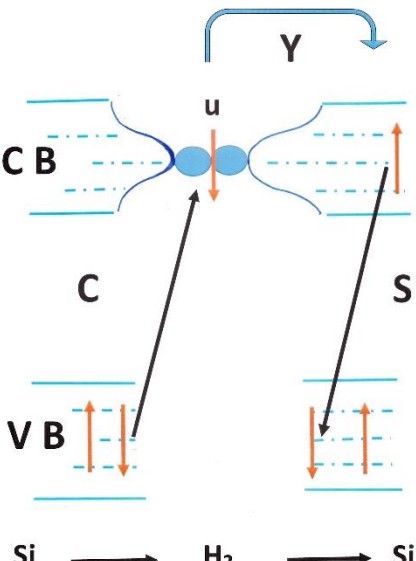

**Figure 13.** CY and SCY Conversion Processes of Molecular Hydrogen diluted in Silicon. In the depicted Singlet path, the repulsion **C** pulls out one valence electron k towards the molecular antibonding state **u**. The spins of the excited **u** and unpaired solid electrons k are opposite. The contact interaction **Y** transfers back to the solid the **u**-electron to κ in the CB, building a magnetic triplet exciton state and inducing a double nuclear and electron spin flip-flop. In the Triplet path, the contact interaction transfers first a valence electron to form a triplet and jumps in the CB by electron repulsion. For all channels, the electron spin–orbit coupling Λ relaxes these magnetic excitations throughout the solid. The efficiency of such processes depends on the possible dressing of the molecular antibonding excited state u by the solid conduction band, as well as on the spin-orbit strength.

Consequently, an important property of the fullerene cage is its magnetic shielding. The subject is to increase the nanocage magnetic shielding for memory applications, production and conservation of parahydrogen for very long periods, for instance, medical imaging or particle detections. Note that we only stressed the influence of shortly lived excitons to interpret the $H_2$@Si measurements and found negligible effects of the very small permanent electron transfer to the Si cages. Experiments with an increased concentration of doping impurities would be useful to describe the transition from insulating to doped semi-conductors and to metallic hydrogen catalysis.

### 4.2. Collective Phenomena in Catalyzed Conversion

I used the concept of an electric nanocage to compare the behavior of $H_2$ molecules in front or inside such confined structures. It is necessary to remind and distinguish between different types of structures. Ones pertain to the category of aggregates or super molecules such as $H_2$@$C_{60}$ for which the concept introduced by Van der Walle, of multiple valence bond should be considered [130,146]. Distinctly, 2d and 3d crystal structures offer properties for which collective effects are part of the catalytic mechanisms.

In between disordered solids such as amorphous ones or powders such MOFs or porous polymers such as "PCP", or viscous solvents solutions such as acetone, offer more possibilities to the diluted hydrogen to escape and wander from site to site, diffuse or escape.

In the following, I summarize shortly three types of collective effects: (i) magnetic (ii) electric (iii) thermal. A few magnetic catalysts are exceptionally active to convert hydrogen and they are used in industrial applications. My interpretation relied on the

emission of spin-waves that transfer the hydrogen rotational momenta and energy to the catalyst. Magnetic catalysts are still difficult to observe by optical means; for instance, when 3d ions were inserted in the MOF-74 samples the conversion rate was too fast to be followed by IR absorption [97].

Note that a few supra-conductors present conversion patterns but their mechanisms have not been explained until now. For dielectric ionic catalysts, I considered two possible collective effects: one is an emission of electron-hole pairs of different forms for either metallic, or semi-conductors or insulating substrates. Another collective effect is thermal and considers the heat exchanges between the hydrogen molecules and the thermal bath.

### 4.2.1. Ferromagnetic Catalysis by Magnon Emission

I discovered very soon the possible inference of magnetic excitations in the catalytic mechanism of the hydrogen conversion. First in 1968 when I compared the competition between the electron spin fluctuations of paramagnetic surfaces and the thermal motion of the catalytic partners in the OSU group of Pr. J. Korringa [16]. Roughly, when two dynamics compete the fastest dominates (the various correlation times compete through their inverse). Thus, I ascertained that fast electron spin relaxations might lead to the conversion for some paramagnetic surfaces at low temperatures. For magnetically ordered samples, I studied in 1970-73 the possible transfers between the molecular rotational momenta and the spin-wave spectrum of the catalyst, in the Laboratory of Pr. J.L. Motchane. In 1986, S. Sugano and I analyzed the effect of surface–molecule electron exchanges for hydrogen molecules adsorbed on chromia-alumina catalysts.

As represented in Figure 2, the conversion rate rises sharply as the catalyst orders. I suggested in 1970 that the conversion energy reported on the magnetically dense catalysts might have been transferred to the magnetic system and not first, as usually assumed, to the thermal bath by either the phonons or the molecular motion. I labelled such a process: "Resonant Exchange of Energy" between the adsorbed hydrogen molecules and the magnons of the magnetically ordered catalyst [16,17].

Let me describe a simplified pure case of an ortho molecule sitting and vibrating in front of an electron spin located at a surface site of a ferromagnetic sample. The static component of the spin along the magnetization axis creates the inhomogeneous magnetic field necessary to break the o-p selection rules, whereas the dynamic transverse component rotates at the magnon frequencies $\omega_q$. If one of the magnon frequency $\omega_q$ coincides with the ortho-para main frequency: $\omega_q = \omega_{op}$, the molecule can relax to its para-state by emitting a spin-wave of that frequency. In the original paper, the transfer was formulated in terms of a spectral density: $J(\omega_{op}) = \int A(\omega)B(\omega_{op} - \omega)d\omega$, where $A(\omega) = n(\omega)$. $f(\omega)$ includes the magnon density of states $f(\omega)$ and the Bose distribution $n(\omega)$ at the temperature T. The function $B(\omega)$ is the Fourier transform of the time-dependent correlation function. For the simple model of exponential correlation times for the molecular motion, a Lorentzian form is founded for B: $B(\omega_{op} - \omega) = \frac{2\tau}{1+(\omega_{op}-\omega)^2\tau^2}$. It is apparent that for $\omega_q = \omega_{op}$, the spectral density is strongly peaked by a factor of the order $\omega_{op}^2\tau^2$, at least. The magnetic catalysts such as $Fe(OH)_3$, $Cr(OH)_3$, $Ni(OH)_2$, whose molecular fields are about 100 kG, belong to the category of weak ferromagnets such as ferrimagnetic ones and have a high probability to possess a spin-wave in the resonant region $\omega_q = \omega_{op}$. They were thus assumed to belong to the class of magneto-catalysis effectiveness [16–20]. On the other hand, strong ferromagnets such as $Fe_3O_4$, or pure Ni characterized by optical magnons or paramagnets have a lower probability to exhibit a spin-wave of high density around the resonant condition. These interpretations were considered later by K.G. Petzinger and D.J. Scalapino [21] and by Y. Ishii and S. Sugano who studied different models of molecule–magnon exchanges [147].

### 4.2.2. Excitonic Dissipations in Dielectric Conversion

For noble metals, I interpreted the EELS experiments, in collaboration with Pr. A. Yoshimori in 1991, by the emission of electron-hole triplet pairs which carry away the ortho-para (o-p) rotational energy as well as the nuclear angular momenta dissipated

through the metal. I considered the virtual charge transfer towards the molecule, and its consequent electron-hole pairs emission. My model led to a conversion time on Ag(111) of about 1–10 min [35–39] and not hours or days as previously calculated [143]. That time and the rate enhancement were confirmed by REMPI in 2003 in the Tokyo group of Pr. K. Fukutani [6,41,42]. Later on, in the Brazilian UFSCar group, I studied between 2013 and 2019 the possible emission of excitons associated with virtual or real charge transfers between $H_2$ and dielectric insulators. I compared these high-frequency electronic excitations to low-frequency nuclear ones [129–131].

More recently, I extended the quantum formalism to relate the dynamical hyperfine information, brought by the conversion measures, to the « Solid » band structure [129]. To summarize our theoretical description of $H_2$ molecules in a solid host, we remark that the molecular "ungerade" and "antibonding" states are admixed to the conduction band, allowing a partial delocalization of a molecular excited electron inside the solid band and inversely a partial localization of a conduction electron on the molecular edge, as illustrated in the Figure 14.

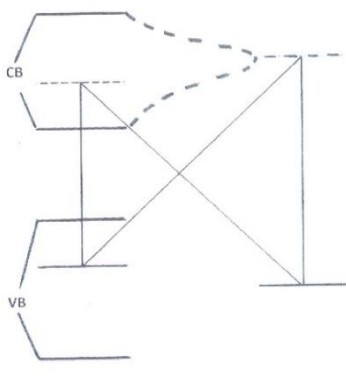

**Figure 14.** Bonding–Antibonding and even–odd transitions in H2@Solids systems. At zero order, the solid electron basis is composed of a full valence band ($\xi \in$ VB) and an empty conduction one ($\eta \in$ CB). The molecular one has a full bonding state: g = $\sigma_g$(1 s)and an empty antibonding one: u = $\sigma_u$*(1 s). By solid–molecule perturbations, four bonding–antibonding mixings are operating and involve excitation energies: (i) interbands: $\varepsilon_{\xi\eta}$, (ii) purely molecular: $\varepsilon_{gu}$, (iii) charge transfer towards the solid: $\varepsilon_{g\eta}$, (iv) charge transfer towards the molecule: $\varepsilon_{\xi u}$. (The molecular bonding energy: $\varepsilon_g$ lies deeper than represented, $\varepsilon_g \approx -15.8$ eV). The dashed curve represents the "dressing" of the molecular antibonding u state by the conduction electrons.

The electron repulsion performs the molecular rotation transitions by excitation of higher orbital momenta states, while their delocalization enhances their contact with the hydrogen nuclear spins. Quite often it appears difficult to distinguish between the charge transfer and the exchange processes, probably both contribute to the conversion rate and their relative influence is a matter of the particular cage characteristics. Similarly, it is still difficult to appreciate the relative importance of the electronic or the nuclear process which is a function of the connection between the cage that incorporates a molecule and the remaining solid.

4.2.3. Thermal Accommodations in Nano-Cages

The question of the thermal accommodation of the hydrogen with a solid is related to the molecular statistical patterns and more precisely on the internal partition functions. Let us distinguish between three situations (i) the most usual one is to consider that the hydrogen molecule energy spectrum remains unchanged in front of the solid and at

thermal equilibrium with it, (ii) the second changes the energy spectrum as a result of the electromagnetic interactions with the solid but keeps the thermal equilibrium (iii) the third considers the non-equilibrium nature of an arbitrary mixture of hydrogen spin isomers. (i) The statistical thermodynamic description of a hydrogen sample has recently received significant attention and is related to the models of the equation of state. The interested reader will find important studies in the literature [148–152], differing in the way the nuclei-electrons couple together. Note that they all rely on the Born–Oppenheimer approximation and the resulting separation of the degrees of freedom. Such validity should be checked carefully in the case of strong adsorptions at the alloying limit. (ii) A different equilibrium might be obtained for mixtures submitted to an external electric potential, as for hydrogen adsorbed on a surface or diluted in a solid. In such cases, the rotational energies, different from the gas values, might also be split. I shall report the interested reader to the ancient study of T.B. Mc Rury and J.R. Sams [153], and references therein, in particular to the pioneering works of Y.L. Sandler and A.A. Evett. Usually, two extreme cases are considered, the one where the molecular axis is aligned along the surface normal and the other consisting in a rotation parallel to the surface and perpendicular to the normal (or to the molecule–catalytic site direction). For the equilibrated spin isomers mixtures, it is common to define a separation coefficient s(T) that measures the relative variation of equilibrium composition: $s(T) = \rho(s,T)/\rho(T)$, where $\rho(s,T)$ and $]\rho(T)$ are the ortho-para equilibrium ratios, respectively in the modified phase and in the pure gas. (iii) The previous studies are all devoted to special ortho-para mixtures, namely equilibrium ones: **eH$_2$** at the temperature T, the normal one **nH$_2$** is equilibrated at the temperature of 300 K and the para one: **pH$_2$**, equilibrated at 0 K. Such statistical descriptions are not valid for an arbitrary proportion of the ortho and para varieties. In Section 1.2, it was noticed that the conversion process corresponds to a change of one variety into another and thus that the conversion "creates" a new entropy corresponding to a change of the mixing entropy. It is a peculiar behavior of the hydrogen fluid to be characterized by a molecular discernability change.

The usual picture of a molecule, being trapped in the physisorption well, is that it vibrates around an average distance, where the long-range dispersive attraction counterbalances the short-range quantum repulsion. Infrared measurements have registered, broad translational lines around 4215 cm$^{-1}$ in addition to internal vibrations. Subtracting the pure vibrational frequency, the authors estimate the translational frequency to be 123 cm$^{-1}$.

These broader bands of the weak coherent part seem to arise from transitions in which the molecules also increase their center of mass translational motion, as if the molecules were "rattling back and forth" within their cage. When thermal equilibrium establishes between the solid sample and the gaseous hydrogen, the conversion heat is evacuated by the thermostat without affecting its temperature.

In hydrogen catalysis, the collective effects manifest themselves in the way they furnish or evacuate the o-p transition energy. The solid–molecule electron system feels the difference of temperatures between the molecular nuclei and the thermal bath, and consequently transfers the rotational angular momenta to the vibration and translation ones, catalyst phonons or molecular scatterings, through their mutual Coulomb interactions. The o-p conversion requires the energy dissipation process because the para-state is energetically lower than the ortho state. Therefore, the role of phonons in energy relaxation should be considered. In order to discuss the influence of the solid phonons in the conversion process, let me return to the interpretation of the steep temperature dependence of the conversion rate on the ASW surfaces at very low temperature, as measured in 2013-6 by the Japanese teams [154,155].

The phonon process generally shows a temperature dependence and thus may be able to explain the observed steep temperature dependence. Among the phonon processes for spin-lattice relaxation, the obtained temperature dependence below 14 K is best fitted by the equation for the two-phonon Raman process, whose rate is proportional to T$^7$, when the temperature is sufficiently lower than the Debye temperature of the solid [79]. An inelastic neutron-scattering experiment indicated that the acoustic phonon modes

of vapor-deposited ASW at 50 K cover energies below approximately 23 meV with two very broad peaks centered at approximately 7 and 17 meV. Such a fitting reproduces the experimental data below 12 K quite well, but fails when including the plots at 14 and 16 K. Above 12 K, the conversion rate deviates clearly from a power law. I interpreted that deviation by suggesting that the conversion energy might be first transferred to the molecular motion before being evacuated by the catalyst phonons. The suggested **SCY** rate fit the experimental ones in magnitude and steep temperature dependence, as well as the following decrease. Fundamental discussion focuses on the relative share of the rotational energy dissipation by the solid phonons or by the electromagnetic nature of the process [131].

## 5. Industrial Prospects and Concluding Comments

That chapter is devoted to prospective studies. The first application illustrates the relationship between purity and imagery in medical nuclear imaging devices. The second concerns the relation between memory and dating illustrated by recent researches about the history of the hydrogen molecular formation in interstellar dust (and its possible memory). At last, a few industrial applications such as the hydrogen energy storage and liquefaction procedures, which are becoming essential now to face the climate challenge, illustrate the macroscopic character and singularity of the hydrogen nuclear spin isomers. The chapter is concluded by a short summary, and discussion on the past and possible orientations in future hyperfine researches.

### 5.1. Memory and Imagery

The two main properties of molecular hydrogen are the basic factors that link the memory and imagery applications: the first one being its lightness and the second its quantum structure. The small inertia momentum induces large energy separation between the ortho and para states and the fermion character of the protons links the nuclear spins to the rotational states. For instance, the first o-p energy corresponds to a Terahertz frequency about $10^5$ larger than the usual nuclear magnetic ones [3].

### 5.1.1. Purity and Imagery

The very long lifetimes of the hydrogen nuclear spin isomers have applications in various contexts for research or industrial purposes. The preparation of hydrogen molecules in the "pure" para-state (J = I = 0) and the following transfer of the nuclear polarization to carbon nuclei were at the origin of giant signals in medical NMR Imagery [69,74–76,156]. The method of molecular surgery developed by Turro's group allows improving the magnetic shielding and the long-term conservation of the para variety [105–117,157,158]. For a long time, physicists and chemists have tried to separate the ortho and para varieties.

One of the first and successful attempts was reported in 1958 when Cunningham and Johnston demonstrated that the ortho molecules were preferentially adsorbed on alumina surfaces and by passing hydrogen gas through a chromatographic column obtained hydrogen-enriched in 99% of the ortho manifold [159]. Such a method is still used nowadays, as by K. Fukutani et al. who prepared a non-equilibrium hydrogen mixture by chromatography before its adsorption on silver and water samples. The LID (Light Induced Drift) radiative methods that have been successful in the separation of $CH_3F$ spin isomers are still inoperative for separation of the hydrogen isomers because of their lack of electric dipolar momentum [57,58].

The catalyzed hydrogen liquefaction is able to prepare 99.98% pure para $H_2$ at very low temperatures but is difficult to keep at higher temperatures [2,15]. However, even such a purity remains still insufficient for para $H_2$ to be used as a nuclear probe and detector for the dark matter DM. Because the dark matter in our galaxy is rare it is not easy to detect. In the case of the pure p-$H_2$ sample, o-$H_2$ signal could be observed as a signature of DM-nucleon interaction but the impurity of o-$H_2$ in the sample needs to be below $10^{-10}$, which is presently unattainable.

A successful method to use the para pure state is to transfer the hydrogen polarization to a different atom. Para-Hydrogen-Induced polarization (PHIP) is a powerful technique for studying hydrogenation reactions in gas and liquid phases [74–76,119,120]. It provides a sensitive readout to investigate the kinetics, selectivity, and nature of catalytically active sites, important in the context of future studies of these reactions on a larger scale and potentially on an industrial scale. Pairwise addition of parahydrogen to the hydrogenation substrate imparts nuclear spin order to reaction products, manifested as enhanced $^1$H NMR signals from the nascent proton sites.

This property has made PHIP a unique tool for practical medical imaging that benefits from the giant NMR signals based on the polarization transfer of para molecules. It leads also to the numerous mechanistic investigations of catalytic reactions, due to the possibility of sensitive intermediates detection and the ability to track hydrogen atoms from the same hydrogen molecule [69,76,121].

### 5.1.2. Memory and Dating

The spin-singlet nature of the para molecules increases the memory of the hydrogen nuclear spins. Their pure state allows storing their nuclear spin order in extended periods. I demonstrated that such an effect occurred in Canet's observation of hydrogen conversion in organic solutions [126]. Long singlet states lifetimes have been observed when they are prevented from mixing with triplet states. By using magnetic field cycling, slow singlet relaxation beyond the spin-lattice relaxation times were observed at room temperature [160–163]. These technics facilitate the development of nuclear spin hyperpolarization methods.

The combination of hydrogen spin isomers mixtures in non-thermal proportions, together with "site-specific" spectroscopy, has brought important information on surfaces [5,6,144] and is also used to follow the transient states of chemical reactions [121]. In astrophysics, the ortho-para ratio (OPR) helps to date the formation of interstellar clouds [154]. It characterizes the birth of the hydrogen molecules from independent atoms, with the appearance of the two nuclear spin isomers, as used in the dilution of hydrogen in silicon structures, described in Section 3.1.2.

It is generally assumed that when two hydrogen atoms form a molecule, their ortho-para ratio (OPR) = 3, because the ortho variety has a threefold nuclear spin degeneracy, assuming an equal probability of the relative spin orientations. However, that argument should be carefully investigated in view of molecular spin-rotational possible couplings. In particular, when a hydrogen molecule is formed on a cold surface with such a ratio (denoted normal hydrogen), it corresponds to a "hot" proportion. If the nuclear temperature is quite different from the surface one the coupled system is submitted to a heat flux.

The tunneling hydrogenation is one of the important features that enhances the molecular formation on cryogenic dust surfaces. On a dust surface, adatoms need to migrate for a long distance to encounter their reaction partners and reaction rates are often limited by diffusion. Microscopic techniques such as STM, FEM, and AFM often used for monitoring adsorbates are not powerful for detecting H atoms, and pure water ice is not an electrical conductor.

The recombination reaction itself would proceed immediately even at approximately 10 K once the atoms encounter each other, because radical–radical association reactions tend to be barrierless. $H_2O$, $H_2CO$, and $CH_3OH$ molecules have been found abundantly as solids in the ice mantles that cover the cosmic dust [154]. The surface reactions on silicate dust start with $H_2O$ formation as the main component of the ice mantle.

Once $H_2O$ forms building the first layer of ice on silicate dust, subsequent reactions occur on the water ice. In the series of reactions, the last step of the $O_2$ hydrogenation and the first and third steps of the CO hydrogenation proceeds via tunneling because these reactions have significant activation barriers corresponding to 2500–3500 K and therefore cannot occur thermally at approximately 10 K. Electron fluctuations at these surfaces play an important role in these tunneling processes. However, such a phenomenology is an

intricate matter because the surface–molecule electron exchanges are admixed with the nuclear ones.

The non-equilibrium proportion of the spin isomers induces a difference between the nuclear spin temperature and the electronic one creating an irreversible drift [126].

*5.2. Hydrogen Liquefaction and Storage*

5.2.1. Hydrogen Liquefaction

In the hydrogen liquefaction plants, the feed hydrogen is compressed, mixed with compressed recycled hydrogen and precooled to nearly liquid nitrogen temperature. The product stream passes through a catalytic converter which converts the hydrogen to approximately 45% para concentration [2,3,15]. It is further cooled against cold exhaust streams from the turbines followed by a final stage of cooling and conversion to at least 95% para hydrogen. The ortho-para catalytic conversion is shown as a batchwise procedure; the feed hydrogen is partly converted in a first stage, operating at the temperature of liquid nitrogen, and finally converted in a second stage, operating at the temperature of liquid hydrogen. This process requires less work than one using a single stage of conversion at liquid hydrogen temperature because part of the conversion heat is released at a higher temperature level. The work required for a multiple-stage process is an inverse function of the number of stages until minimum work is obtained with an infinite number of stages. The latter condition can be approximated in a practical way by using continuous converters [164–169] which are characterized by the inclusion of a heat transfer function. The feed stream, in passing through the catalyst bed, is cooled by a counter-current stream of refrigerant hydrogen gas and, in the process, the heat of conversion is transferred continuously at temperature levels that are only a few degrees below the equilibrium temperature. The minimum theoretical work for liquefaction of hydrogen is that required to reversibly remove heat from hydrogen. Several calculated values for the work of liquefaction and liquefaction plus conversion have been reported [15]. To liquefy normal hydrogen reversibly requires 12.09 kJ/g; the conversion to 99.79% para adds 18.1% to the minimum work requirement. The effect of converting to para hydrogen in a stagewise manner is to add up to 35.6% more work when the conversion is carried out in a decreasing number of stages.

Today's market for liquid hydrogen requires considerable storage time between liquefaction and ultimate usage. The most economical approach to this is to produce the liquid hydrogen at a relatively high para concentration using continuous conversion below liquid nitrogen temperature. This minimizes boiloff during the relatively long storage period. The large-scale use of liquid hydrogen as a fuel for jet aircraft presents a different situation.

Utilization of the liquid is more nearly a continuous operation with little need for long-term storage. Liquid hydrogen could possibly be used in less than a day after it was produced. Long-term storage would be used only to take care of possible plant outage that is not covered by multiple train installations. For each initial composition, a breakeven time exists for which the energy cost for conversion equals the energy cost for the vaporized hydrogen. If the hydrogen is used within the breakeven time limit, partial conversion is advantageous with respect to energy consumption.

5.2.2. Hydrogen Storage

Finally, it is interesting to note that a better understanding of the hyperfine measures of $H_2$ in electric cages might contribute to open new research directions related to $H_2$ storage [170–175]. For cryo-adsorptive storage, porous structures such as activated carbons, MOF or zeolites are studied. The general tendency is to look at adsorbents with increasing binding strengths, where each site can accommodate several $H_2$ molecules.

For hydrogen storage, chemisorption bonds hydrogen strongly, much more than the Department of Energy's 40 kJ/mol target. The sorption process in this case is not easily reversible, and so the energy stored cannot be practically extracted to power an automobile. Finding materials that chemisorb more weakly and relinquish their hydrogen

more readily has been a major focus of study. Physisorbent materials produce lower binding energies than desirable, usually less than 10 kJ/mol [95]. It is much easier to extract hydrogen, though in many cases too easy. Many physisorbent materials have little to no hydrogen uptake at room temperature and atmospheric pressure. They require either low temperatures (<100 K) or high pressure (>100 atm), making long-term storage under practical conditions impossible [3,15].

The hope is to find materials with higher binding energies, though not too high, so that they might be able to store efficiently under practical conditions. Another essential requirement is to minimize the mass of all components of the storage system. Given the 9 wt% Department of Energy goal (ratio between the mass of stored hydrogen and mass of storage material) means lightweight, porous material with high surface area. This has historically meant the use of carbon allotropes and zeolites. In the last decade, attention has turned to a new class of materials called Metal-Organic Frameworks (MOFs), which consist of metallic clusters connected by organic links. They feature a tunable structure, allowing for substantial customization of adsorption materials. They exhibit binding energy between 8.3 and 8.8 kJ/mol.

### 5.3. Concluding Comments

The conjunction of new measures, new materials and new concepts have brought an important renewal of the ortho-parahydrogen accommodation and catalytic conversion. The following comments will be divided into three parts: First, recognize the nature of the changes that affected the hydrogen physical catalysis, then question the various research limits and insufficiencies and finally anticipate some future directions.

### 5.3.1. Identification

It appears necessary first to recognize the extension of the recent researches on the hydrogen nuclear spin isomers. The hydrogen conversion renewals are related to basic concepts, measurement methods and materials involved in the hydrogen research and development activities.

#### Concepts

Among the few conceptual understandings that have evolved, the main one concerns the magnetic selection rule. Over the last century, all theoretical and experimental investigations on the hydrogen physical conversion were performed over magnetic catalysts. If we denote by **S** the catalyst magnetic quantum momentum, the resulting o-p transition selection rule can be represented by $\Delta S = 0$. New phenomena involve processes on non-magnetic catalysts and the new transition selection rule becomes $\Delta S = 1$, corresponding to a transfer of spin-orbital momentum to the catalyst. The second domain concerns the scattering conditions between the hydrogen sample and a solid. In the past observations, most conversion measurement mechanisms were registered at the catalyst surface, with molecules sitting in a minimum of the surface electric potential. In porous solids the encapsulated molecules are not necessarily at equilibrium with the catalyst nuclear system, spending more time "off-center" of the cage. Collective phenomena might drive the interacting partners, evolving towards a global equilibration. The third conceptual domain concerns the role of the molecular electronic excited ones. It is true that the excited energies are high, but the electronic coupling interactions also, particularly during the scattering durations. Moreover, the nuclear interactions are so weak that even a small electron admixture produces competing effects. I distinguished and classified the conversion mechanisms in the function of the number of interacting links of the chain that represents the conversion trajectory in the molecular configuration space.

#### Measurements Methods

I shall here distinguish between the time procedures, the frequency ranges of the radiative measure and the experimental conditions of the measurements. Concerning

temporality, progress is major. In the past, most measurements were macroscopic in measuring global isomer concentrations in gas samples and often static. Nowadays the measure is dynamic, and mostly radiative of various frequencies: IR, UV or Radio. The IR measurements of hydrogen in MOF samples realized by S. FitzGerald's team in Oberlin from about 2005, were able to follow the absorption lines of each isomer in real-time almost continuously. Moreover, the molecules adsorbed on different sites were distinguished one from another by their different adsorption frequencies and even interactions between them could be registered by varying the loading of the different sites. The progress registered in the use of radiative methods occurred almost simultaneously and independently. First Raman spectroscopy, neutron and electron beam scatterings applied from 1980 remained insufficiently precise to measure the spin isomers. Later, apart from the IR measurements, another major change appeared in 2003, when K. Fukutani and his Tokyo team could measure the conversion time evolution by the REMPI method, an UV irradiation described in 2.2. The measurements performed for molecules on silver or amorphous solid water allowed decisive progress in the observation of hydrogen molecular formation and reaction on diamagnetic substrates.

The experimental conditions were also valuably extended in many reactions, for example down to very low temperatures in the IR and UV methods; in anticipated preparations of isomer mixtures from plasma or by chromatography; in delayed reconstitutions after laser-induced molecular desorption from Ag or ASW samples. The observation of the reaction path sensitivity over the initial isomer concentration in viscous organic solutions illustrates the irreversible nature of the conversion pattern.

Materials for Hydrogen Conversion Studies

Most magnetic catalysts for o-p conversion considered in the past were oxides. Commercial chromia-alumina is still considered as a reference. P.W. Selwood reported an important number of conversion measurements on supported oxides by dispersing partly filled 3d shells transition ions: on alumina surfaces of area 100–300 $m^2/g$, or 4f sesquioxides supported on lanthana. Numerous magnetic field effects were registered around $10^3$ gauss for 3d samples and at very low fields around 10 gauss for rare earth oxides. For the NBS laboratories that elaborated the most efficient catalysts for industrial purposes, the hydrous oxides of iron and nickel, in particular, were found more active.

Nowadays, the need to increase hydrogen storage and water electrolysis performance has created a drive towards porous and composite structures. We have described the importance of the MOF powders, activated carbons, graphite prepared by pyrolysis, organic solvents and nanocages such as fullerenes. Multiple decked sandwich clusters and nanoparticles of late transition metals are studied as catalysts and electrocatalysts for industrial chemical reactions that produce fuels, convert chemical energy to electricity and clean up pollution devices.

5.3.2. Questions

We might wonder why so many and deep changes that have occurred in that field needed such a long time to be accepted and recognized. I shall survey shortly a few possible reasons: (i) inherent complexity of the reaction mechanism (ii) dispersion of the physico-chemical fields and scopes (iii) socio-economic structuration of the research groups and of the publishing edition. (i) The inherent complexity of the conversion process proceeds from the number of systems interacting with one another, together with a thermal bath and with the radiative measurement. In such a context the overall simplification, brought by the Wigner model in 1933, provided a simple and popular basis for the interpretation even up to now. For example, when P. Avouris observed the disappearance of the orthohydrogen deposited on a silver surface, most colleagues were convinced that the IBM samples were not cleaned enough and some magnetic impurities had remained on their samples. The Chalmers group of S. Andersson interpreted the disappearance of ortho manifolds on noble metals by a preferential adsorption mechanism. The research group of Dr. E.V. Lavrov

published up to 2009 numerous conversion measurements in silicon samples, interpreted by dipolar interactions with hidden magnetic isomers:[75]As, [31]P or [29]Si. Even up to 2013, the conversion rates measured in fullerenes were interpreted in terms of the spatial ranges considered by the Wigner model.

The reason for the slow progress of the o-p conversion proceeds also from its multi-disciplinary character. The nuclear spin isomer studies are at the crossing of a number of scientific fields: surface and solid-state, molecular spectroscopy, magnetism, thermodynamics and irreversible processes, chemical reaction, and more. To be concrete let me illustrate the need for intensive collaborations by an example: the important report on the conversion of hydrogen in a PCP polymer [111]. It required the collaboration of nine experts belonging to eight different institutions. The study required the following steps: sample preparation by chemical synthesis, structural analysis by neutron and X-ray diffraction synchrotron radiation and magnetic susceptibilities measured by a SQUID magnetometer. Hydrogen in the sample was measured by a combination of adsorption isotherm, and the Raman in situ spectroscopy was carried out by an iHR320 spectrometer in the temperature range of 20–77 K under temperature control. The data were registered with high counting statistics, at 35 and 65 K in the $H_2$ adsorbed and desorbed states, accurate charge densities and electrostatic potentials in the pores were determined by combining the maximum entropy method (MEM) and Rietveld refinement.

Moreover, deep changes in the research direction need time and investment. For example, it took 23 years between the first IBM observation of $H_2$ conversion on Ag samples [32] and its experimental confirmation by the ISSS group [41]. A similar delay occurred in my interpretation of the non-magnetic conversion: 23 years were needed to extend the metallic case [35] to the insulating catalyst one [127–129]. The need for rapid publications to obtain financial investment contradicts such long-term research.

### 5.3.3. Anticipation

Nevertheless, progresses related to that review are so important that it becomes urgent to draw a few directions and anticipate possible applications. It is necessary to distinguish first the information gained on new catalysts by the hyperfine measurements, from the direct hydrogen applications.

The new role of hydrogen as a powerful energy carrier drives important progress for storage systems, fuel cell devices, hydrogenation and organic reactions as well as medical imaging, shortly evoked in the last Section 5.1 and Section 5.2. The need for more economical hydrogen liquefaction requires a continuous conversion process during the cooling which would improve the yield by a reduction of practical energy consumption of about 25% when compared to the present successive stagewise technology.

The need of increasing the fundamental investigations, theoretical and experimental, is also particular to the hydrogen field. One of the most important characteristics of the hydrogen molecules proceeds from the peculiar macroscopic effects resulting from their fundamental quantum structure. For example, the conversion energy associated with a nuclear spin "flip", about 170 K in Kelvin unit, affects the liquid storage at 20 K as well as the initial formation of molecules in the interstellar dust. Other examples of progress are developing in the photoelectrochemical cells, one-dimensional semiconductor structures, encapsulation in single-walled carbon nanotubes.

The particular sensitivity of the hyperfine interactions to the fast electron fluctuations of the catalyst is also becoming a domain of investigation in the surface physics, the new composite structures and curved topological catalysts. The fast growth of the "Terahertz" technology is particularly suited to hydrogen physics (the ortho-para fundamental frequency amounts to about 3 Thz!).

However, despite the importance and variety of the new radiative and electronic devices, the measurements and interpretations that have been reported since the turning of the new century are only a first step in the development of a new hydrogen economy.

**Funding:** This research received no external funding.

**Institutional Review Board Statement:** Not applicable.

**Informed Consent Statement:** Not applicable.

**Data Availability Statement:** Not applicable.

**Acknowledgments:** I am particularly indebted to Jan Korringa who initiated me to the theoretical research, to J. Uebersfeld and J.L. Motchane for their collaboration in creating theoretical research groups at the University of Paris, to J.R. Gaines, P. Wigen and the Low Temperature Group for fruitful discussions and contracts at the Ohio State University, to S. Sugano, A. Yoshimori, K. Fukutani and K. Aoki for intensive collaborations and exchanges in the Universities of Osaka, Tokyo and Kyoto, to E. Mirra and F. Ghiglieno who introduced my lectures in the Federal Brasilian Universities and collaborated into my hydrogen researchs.

**Conflicts of Interest:** The author declare no conflict of interest.

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
