# Peer review of "Hydrogen Conversion in Nanocages"

_hydrogen, doi:10.3390/hydrogen2020010_

Round 1

Reviewer 1 Report

The title is enticing, and the word nanocages gives it a flavour of modernity and up-to-date work. Unfortunately, the abstract does not disclose what kind of "conversion" is meant. Conversion has many meanings, in many fields from religion and psychology to technology and applied chemistry. It would be good to tell in the abstract which type of conversion is addressed. I have the feeling that the author is bursting of energy to tell a rich story of H2 and H, the insight the atoms and molecules have offered and a glimpse of the many fascinating facets of their interactions with other atoms and molecules, be they individual particles or structured matter. 
He catches my attention - but I would prefer not to be dumped into an ocean of information without a briefing on what to expect and at least gross directions. Also I see that the author is happy to talk about his field, and that he is highly fluent in English. At his high speed already line 7 gives me a hard jolt: "leaded" as in "leaded petrol", or the wrong past tense of the irregular verb "to lead"? In that case, "led" would have the correct form. I see that I have to be careful not to become swept away with the author's elan, having to expect author-induced strange things along the path. 

So far, most of the abstract belongs into the introduction instead. The new abstract ought to indicate the idea of the manuscript, here the intention and scope of a review of a surely interesting topic, not the importance 
of something that was achieved in 1927 - while the review is to be on matters of this century. Some more self restraint and mental discipline by the author might greatly benefit the prospective reader, who is 
expected to digest this flood of information. 

After the abstract the text quickly accelerates to speed. I suggest to reduce the initial acceleration, by explicitly mentioning some very basic points and entities. Not every reader will have read the earlier reviews, 
and a very short (verbal) introduction on the property of the hydrogen atom that matters here, nuclear spin, and on ortho and para hydrogen molecules and their peculiar features, would help the reader to find some orientation. No doubt much will be told later in the text, but it is sensible with respect to the reader to fetch his/her attention in manageable steps. That would not take much text, but a little more thought on optimum information transfer and processing.  The present introduction actually evolves into the right mode towards its end - the abstract and the first two thirds of the introduction better be modified thoughtfully so that they no longer shock and frighten the reader away from continuing with the intriguing topic. 

line 77: "slitted" ought to read "split" 

line 83, "dipole" needs no special character 

line 106, the use of "differently" seems odd and un-English. Rephrase the sentence to make it better. 

line 132, the sentence beginning with "Mostly" is incomplete, but could be connected to the preceding sentence 
by a comma. 

line 140, "ferric" is neither an oxide nor an element; maybe replace by "iron" and add "of" before "manganese"? 

line 169, the word is "undoubtedly" 

By now the text has become highly readable, so I am dropping my close attention to English spelling and grammar. The examples so far show, however, that it would be good if the author was to spend more scrutiny on this aspect in the first part of the manuscript. He does a very good job in this section pointing out in passing why ortho-para conversion really is important in practical industrial applications. 

lines 200 and 201: there is a LaTeX misinterpretation of some coding; I see a Chinese character inside the equations. 

The review covers very many examples in a well digestable way. I very much appreciate the way it has been written, except for the first few pages as explained above. 

Refs 7 and 9 change to "Name first, Initials next" from the majority used otherwise. Unfortunately, this minority is closer to MDPI style than the majority. 

Ref 7, a letter "c" missing in a German word, "paramagnetische" would be correct

In conclusion, this is an excellent review for "Hydrogen", but the shortcomings in the early part, which may have been edited for the purpose by somebody who is not the main author, need a revision. 

Author Response

I have appreciated your valuable comments (and humor), and corrected my manuscript along your recommendations.

First the main changes: A new abstract and two additional chapters are introduced in the revised manuscript.

The introduction has been slightly expanded.

In chapter 1, I have regrouped the spin isomers fundamental properties as well as the conversion measures of the past century. In particular I have introduced a new section 1.2 devoted to the thermal properties of the hydrogen rotational system, to explain both the macroscopic isomer property and the non-equilibrium  character of an arbitrary hydrogen sample. That formulation is original. Usual ones are relative to equilibrium mixtures. Unfortunately, I had to introduce a few formula which makes the narration slightly heavier. But I cannot found a more fundamental and elementary characterization of the subject. The non-interested reader might skip that section.

Chapters 2 and 3 are almost unchanged, Chapter 4 also, except that

- I have enlarged the section 4.2.3, in extending its introduction and complementing the thermal introduction of 1.2.

- the Industrial applications have been transferred to a last Paragraph 5, where I have added a few concluding comments (as asked by another referee) to synthetize and discuss a few important achievements of the last 20 years.

The New Abstract is the following :

Abstract: Hydrogen molecules exist in the form of two distinct isomers that can be interconverted by a physical catalysis. These ortho and para forms have different thermodynamical properties. During the last century the catalysts developed, to convert hydrogen from one form to another, in laboratories and industries were magnetic and the interpretations relied on magnetic dipolar interactions. The variety concentration of a sample and the conversion rates induced by a catalytic action were mostly measured by thermal methods related to the diffusion of the o-p reaction heat. At the turning of the new century the nature of the studied catalysts, the type of measures and motivations have completely changed. Catalysts now investigated are non-magnetic and new spectroscopic measurements have been developed. After a fast survey of the past studies, the review details the spectroscopic methods, emphasizing their originalities, performances and refinements: how Infra-Red measurements characterize the catalytic sites and follow the conversion in real time, Ultra-Violet irradiations explore the electronic nature of the reaction and hyper-frequencies driving the nuclear spins. The new catalysts, metallic or insulating, are detailed to display the operating electronic structure. New electromagnetic mechanisms, involving energy and momenta transfers, are discovered providing a classification frame for the newly observed reactions.

Minor Corrections

Line 7 corrected

line 77: "slitted" ought to read "split" corrected

line 83, "dipole" needs no special character corrected

line 106, the use of "differently" seems odd and un-English. Rephrase the sentence to make it better. corrected

line 132, the sentence beginning with "Mostly" is incomplete, but could be connected to the preceding sentence by a comma. corrected

line 140, "ferric" is neither an oxide nor an element; maybe replace by "iron" and add "of" before "manganese"? corrected

line 169, the word is "undoubtedly" corrected

lines 200 and 201: there is a LaTeX misinterpretation of some coding; I see a Chinese character inside the equations. corrected

Refs 7 and 9 change to "Name first, Initials next" from the majority used otherwise. Unfortunately, this minority is closer to MDPI style than the majority. corrected

Ref 7, a letter "c" missing in a German word, "paramagnetische" would be correct corrected

I thank you again for your contribution to improve the clarity of the manuscript.

Best Regards,

Reviewer 2 Report

The abstract is presented incoherently and must be rewritten so that at least it clearly expresses the objectives of the paper.
The work is very difficult to follow due to a poor arrangement and cumbersome expressions of the author, with many problems of technique, grammar, and spelling. The notions are generally correct but are given mixed like an amalgam, the results are not donated correctly and clearly being chaotically exposed in the paper. The work is an interesting one but poorly exposed.
Many figures are unclear and should be replaced with higher resolution ones.
Although the bibliography is thick (175 references) many important works in this vast field are not mentioned, the bibliography being too full of works of the author or his colleagues, or his knowledge.
Many quotations are not given in the order in which they appeared, but later, the latter does not seem to have been used in the text.
It was normal to have a discussion section at the end.
The Conclusions section is missing.
Missing statement of ethics.

Author Response

The Abstract has been rewritten. Here the new abstract:

Abstract: Hydrogen molecules exist in the form of two distinct isomers that can be interconverted by a physical catalysis. These ortho and para forms have different thermodynamical properties. During the last century the catalysts developed, to convert hydrogen from one form to another, in laboratories and industries were magnetic and the interpretations relied on magnetic dipolar interactions. The variety concentration of a sample and the conversion rates induced by a catalytic action were mostly measured by thermal methods related to the diffusion of the o-p reaction heat. At the turning of the new century the nature of the studied catalysts, the type of measures and motivations have completely changed. Catalysts now investigated are non-magnetic and new spectroscopic measurements have been developed. After a fast survey of the past studies, the review details the spectroscopic methods, emphasizing their originalities, performances and refinements: how Infra-Red measurements characterize the catalytic sites and follow the conversion in real time, Ultra-Violet irradiations explore the electronic nature of the reaction and hyper-frequencies driving the nuclear spins. The new catalysts, metallic or insulating, are detailed to display the operating electronic structure. New electromagnetic mechanisms, involving energy and momenta transfers, are discovered providing a classification frame for the newly observed reactions.

The work is very difficult to follow due to a poor arrangement and cumbersome expressions of the author, with many problems of technique, grammar, and spelling. The notions are generally correct but are given mixed like an amalgam, the results are not donated correctly and clearly being chaotically exposed in the paper. The work is an interesting one but poorly exposed.

I’m sorry you did not appreciate the arrangement. I cannot agree with your feeling of a chaotic exposure. I have carefully organized the content along new concepts. However, I have tried to improve the overall manuscript in increasing the first and last chapters. You will find the new table of content at the end of the manuscript. I hope it will help the reader to understand the logics of the exposure.

In chapter 1, I have regrouped the spin isomers fundamental properties as well as the conversion measures of the past century. In particular I have introduced a new section devoted to the thermal properties of the hydrogen rotational system, to explain both the macroscopic isomer property and the non-equilibrium  character of an arbitrary hydrogen sample.

Chapters 2 and 3 are almost unchanged, Chapter 4 also, except that I have expanded the section 4.2.3. and transferred the Industrial applications to the last chapter 5, where I added a few concluding comments to synthetize and discuss a few important achievements of the last 20 years.

I have redrawn a few figures and slightly expanded the bibliography to 187 references.

I am really astonished by your remark that “many important works in this vast field are not mentioned”,

I shall be very grateful to you if you could send me the missing references you are judging important, first because the subject is very important to me and second I shall be happy to include them in the bibliography.

 I am really astonished by your remark that “ the bibliography being too full of works of the author or his colleagues, or his knowledge”. Of course, I cannot include what I don’t know.  I’m very curious to understand your thinking and shall be very grateful to you if you could send me the references you are judging unimportant.

I am really astonished by your remark that “Many quotations are not given in the order in which they appeared, but later, the latter does not seem to have been used in the text.” I have checked carefully the manuscript and found that the quotations are given in the order in which they appear.
It was normal to have a discussion section at the end. I have introduced a Conclusion.

I shall include a statement of ethics.

Best Regards,

Reviewer 3 Report

The paper is a review on experimental methods, diagnostics and theory on the ortho-para conversion of hydrogen molecules. The paper is well written and give a complete overview on the subject. While the description of experimental part is very clear also for not-expert in the field, some more details should be provided to understand the theoretical approaches. 

Regarding the problems related to thermodynamic properties, these two papers should be cited, discussing the calculation of thermodynamic properties of the ortho-para mixture and the definition of the partition function for a low temperature hydrogen.  

[1] A. Popovas and U. G. Jørgensen, Improved partition functions and thermodynamic quantities for normal, equilibrium, and ortho and para molecular hydrogen, A&A 595, A130 (2016)

[2] G Colonna, A D'Angola, M Capitelli, Statistical thermodynamic description of H2 molecules in normal ortho/para mixture, international journal of hydrogen energy 37 (12), 9656-9668, 2012

Author Response

I have appreciated your comments and corrected my manuscript along your recommendations.

In order to take into account your remarks I have expanded the introduction of the Chapters 1 and 4.

 Following your wish to increase the theoretical exposure and in particular the thermodynamic properties  introduced, I have introduced a new section 1.2 devoted to the thermal properties of the hydrogen rotational system, to explain both the macroscopic isomer property and the non-equilibrium  character of an arbitrary hydrogen sample. That formulation is original. Usual ones are relative to equilibrium mixtures. I extended the bibliography and included the interesting references you suggested.

Chapters 2 and 3 are almost unchanged, Chapter 4 also, except that

-  I have enlarged the section 4.2.3 to complement the thermal introduction of 1.2.

- The Industrial applications have been transferred to a last Paragraph 5, where I added a few concluding comments (as asked by another referee) to synthetize and discuss a few important achievements of the last 20 years.

Best Regards,

Round 2

Reviewer 1 Report

The revision has been excellent. This is a paper to keep within reach.

Compliments to the author!

One tiny note:

In line 1892, the unit ought to be THz (not Thz).

Author Response

thank you for your comments

Reviewer 2 Report

There are still some figures that need to be increased in resolution in order to become clearer.
There are still many self-citations at the general level of the work.

Author Response

thank you, the figures have been revised.